# The electrogenicity of the Na+/K+-ATPase poses challenges for computation in highly active spiking cells

**Liz Weerdmeester[1,2], Jan-Hendrik Schleimer[1,2], Susanne Schreiber[1,2]\***

[1]Institute for Theoretical Biology, Department of Biology, Humboldt-Universität zu Berlin, Berlin, Germany; [2]Bernstein Center for Computational Neuroscience Berlin, Berlin, Germany

## eLife Assessment

This **important** study provides new insights into the lesser-known effects of the sodium-potassium pump on how nerve cells process signals, particularly in highly active cells like those of weakly electric fish. The computational methods used to establish the claims in this work are **compelling** and can be used as a starting point for further studies.

**\*For correspondence:**
s.schreiber@hu-berlin.de

**Abstract** The evolution of the Na+/K+-ATPase laid the foundation for ion homeostasis and electrical signaling. While not required for restoration of ionic gradients, the electrogenicity of the pump (resulting from its 3:2 stoichiometry) is useful to prevent runaway activity. As we show here, electrogenicity could also come with disadvantageous side effects: (1) an activity-dependent shift in a cell's baseline firing and (2) interference with computation, disturbing network entrainment when inputs change strongly. We exemplify these generic effects in a mathematical model of the weakly electric fish electrocyte, which spikes at hundreds of Hz and is exposed to abrupt rate changes when producing behaviorally relevant communication signals. We discuss biophysical strategies that may allow cells to mitigate the consequences of electrogenicity at additional metabolic cost and postulate an interesting role for a voltage dependence of the Na+/K+-ATPase. Our work shows that the pump's electrogenicity can open an additional axis of vulnerability that may play a role in brain disease.

## Introduction

The evolution of P-type ATPases in ancestral methanogenic archaea (*Lane and Martin, 2012*; *Dibrova et al., 2015*) also laid the foundation for the energetics of energy-intensive signaling tissues like the metazoan nervous systems billions of years later (*Palmgren, 2023*). In particular, one of the ATPases, the Na+/K+-pump, plays a significant role in charging the batteries required to operate the nervous systems that the motile life of all multicellular animals is so reliant on. The Na+/K+-pump exchanges intracellular sodium for extracellular potassium ions in a 3:2 ratio and thereby generates a net outward current. This electrogenic property of the pump appears not only as a useful exaptation for osmoregulation in eukaryotes (*Stein, 1995*), but also invokes activity-dependent changes in nerve cell excitability (*Contreras et al., 2021*), mediating hyperpolarization after repetitive stimulation and firing-rate adaptation (*Thomas, 1972*; *Ritchie and Straub, 1957*; *Connelly, 1959*; *Nakajima and Onodera, 1969*; *Sokolove and Cooke, 1971*; *Sawczuk et al., 1997*; *Gustafsson and Wigström, 1983*). These mechanisms are in turn exploited in specific neuronal encoding paradigms (*Arganda et al., 2007*; *Forrest, 2014*; *Zylbertal et al., 2017b*; *Megwa et al., 2023*; *Benda et al., 2005*; *Benda, 2021*),

cell-intrinsic bursting dynamics (*Kueh et al., 2016*; *Megwa et al., 2023*; *Zylbertal et al., 2017a*; *Behbood et al., 2023*), and accelerated ion homeostasis (*Gulledge et al., 2013*; *Morotti et al., 2016*; *Niemeyer et al., 2021*) and thus pose an example of jury rigging in evolution. In the study at hand, we show that for nerve and muscle cells that need to be tonically active for long stretches of time (on the order of minutes to hours), the electrogenicity of the pump can have further, less explored consequences requiring special adaptations of the biophysical components that underlie sustained electrical activity.

Generally, it is assumed that the Na$^+$/K$^+$-ATPase instantaneously restores ionic gradients, ensuring the robustness of electrical signaling. Furthermore, it is assumed that the net current that is produced by pump activity has limited effects on spiking activity. Under these assumptions, generally, pump currents are not explicitly modeled and reversal potentials are kept constant at all times. At first glance, this seems a reasonable pragmatic approach both for the interpretation of experimental data as well as for computational models of neural dynamics. Accordingly, relatively little attention has been given to the interference between electrogenic pumping and neuronal voltage dynamics, a trend reinforced by the success of models capturing neural dynamics without pump currents, solely based on fixed ion reversal potentials, such as the classical Hodgkin-Huxley model (*Hodgkin et al., 1952*).

Here, we demonstrate that, contrary to this notion, electrogenic Na$^+$/K$^+$-ATPases can exert a significant direct impact on computational properties of highly active excitable cells, and that Na$^+$/K$^+$-ATPase electrogenicity could pose a challenge for robust spike-based signaling. For strongly active cells operating at high firing rates, additional mechanisms balancing out the pump's effect on computation may be required, imposing extra costs on cell signaling. We present these effects in a conductance-based computational model of an excitable cell, which was extended to include dynamic ion concentrations and Na$^+$/K$^+$-ATPases. To isolate the investigated effects of the pump current and support the generalizability of this study to other cell types, the model includes only classic conductance-based sodium and potassium channels. The only active transporter is the Na$^+$/K$^+$-ATPase, which is responsible for maintaining ionic homeostasis. Modulation of Na$^+$/K$^+$-ATPase activity occurs via its sensitivity to intracellular sodium and extracellular potassium concentration.

While, due to their generic nature, the described mechanisms may pose challenges for any excitable cell that has to rely on electrogenic pumps, we here showcase them in the electrocyte of the weakly electric fish (*Joos et al., 2018*), chosen because of its persistently high firing rates permanently exceeding hundreds of Hz and its, consequently, significant energetic demand. Electrocytes are the cells that make up the electric organ (EO), creating the electric field in the animal's environment that is vital for communication (*Dunlap et al., 2017*). Electrocyte firing rates, and thus the frequency of the electric organ discharge (EODf), are key for the animal's survival: the frequency of the resulting oscillating weakly electric field can be sensed by other individuals through electroreceptor afferents (*Bullock, 1982*) and spans a range of 400 Hz across individuals (*Hopkins, 1974*). It constitutes the primary signal transmitting information about sex and hierarchy and is also used in intraspecific communication. Due to the high-frequency spiking activity and thus high ATP requirement of electrocytes (*Attwell and Laughlin, 2001*; *Salazar et al., 2013*), the EOD cannot only be expected to have been under a severe evolutionary pressure for energetic efficiency, it also exposes the cells to relatively strong electrogenic pump currents that alter cell excitability. The activity-dependent pump currents thus directly influence electrocyte firing rates, as we argue here, complicating the precise regulation of the excitable cells' activity.

Our model suggests two major effects of electrogenic pumps on computation in highly active excitable cells: (1) a significant shift in a cell's baseline activity requiring compensation and (2) strong computational side effects of electrogenic pumping in the presence of functionally relevant input changes. While the first effect is intuitive, as electrogenic pumping permanently contributes a hyperpolarizing current that requires a compensation to keep firing-rate set points, the second effect is less so. In particular, it can result in unexpected rate changes up to a complete silencing of spiking activity in cases where a drastic firing increase was required and induce spontaneous activity where silencing was required. Even when effects are milder, causing only graded firing-rate changes but not silencing, the pump's electrogenicity could interfere with cellular computation when spike timing is to be synchronized across excitable cells. As we show, entrainment of electrocytes by their pacemaker neurons can be disrupted in these cases, weakening the EOD and presumably impairing the fitness

of weakly electric fish. Based on the identified mechanisms, we suggest a diverse set of biophysical options that may permanently mitigate the side effects of pump electrogenicity in both cases.

To do so, we first isolate the effects of pump currents on cell excitability with constant stimulation. We then demonstrate the impact of pump currents in a simple network context, when an electrocyte is driven by a pacemaker neuron. We outline the consequences of electrogenic pumping for two behaviorally relevant signals in weakly electric fish: so-called chirps (i.e. short interruptions in the EOD) and frequency rises (transient frequency sweeps). Finally, we discuss how a voltage dependence of the $Na^+/K^+$-ATPase, compared to a pump that adapts its rate exclusively as a function of bulk ionic concentrations and hence only on long timescales, can be useful in alleviating some of the perhaps detrimental effects of electrogenicity on cellular dynamics.

## Results

The electrogenic property of the $Na^+/K^+$-ATPase plays a role in the osmoregulation of single eukaryotic cells (*Stein, 1995*) as well as, on the organismal level, in marine osmoregulatory organs (*Chung and Lin, 2006*). In metazoan nerve cells, the $Na^+/K^+$-pump restores the cross-membrane ion gradients that generate resting and action potentials, whilst its electrogenicity induces a hyperpolarizing membrane current proportional to its pumping activity. The pump rate's concentration dependence creates a negative feedback loop, leading to an increase in the hyperpolarizing current if nerve cells discharge at higher frequencies where more pumping is required (*Thomas, 1972*; *Ritchie and Straub, 1957*; *Connelly, 1959*; *Nakajima and Onodera, 1969*; *Sokolove and Cooke, 1971*; *Sawczuk et al., 1997*; *Gustafsson and Wigström, 1983*). This feedback loop could thus reduce, terminate, or prevent overactivation. On the other hand, for nerve and muscle cells that need to be tonically active for long stretches of time (on the order of minutes to hours), the electrogenicity could pose substantial challenges, and its consequences may need to be balanced by additional mechanisms, the latter of which further restrict energetic efficiency, as we demonstrate in the following.

### $Na^+/K^+$-ATPase affects tonic spiking

Most computational models of excitable cells follow the principles of conductance-based models for action potential (AP) generation derived by Hodgkin and Huxley (*Meunier and Segev, 2002*). Accordingly, they assume that the compensation of the transmembrane ion currents is carried out perfectly so that reversal potentials remain constant and that the hyperpolarizing current introduced by the pump is negligible (*Hodgkin et al., 1952*). An explicit inclusion of the pump into mathematical models of such cells is hence not required. To challenge this view for highly active cells, we consider the following scenario: Let's assume that an electrogenic $Na^+/K^+$-ATPase is added to the Hodgkin-Huxley type spiking mechanism to compensate ion flow across the membrane. Specifically, we illustrate the effects of electrogenic pumps in an experimentally well-constrained model of the tonically active electrocyte of weakly electric fish (*Joos et al., 2018*; *Figure 1A*). For simplicity, the $Na^+/K^+$-ATPase in this first case is assumed to only depend on intra- and extracellular ion concentrations as in *Hübel et al., 2014*; *Kueh et al., 2016*, and ATP levels are assumed to be sufficiently high not to impair $Na^+/K^+$-ATPase activity. Specifically, a voltage dependence of the $Na^+/K^+$-ATPase (described in some previous studies, see for example; *Weer et al., 1988*; *Wuddel and Apell, 1995*; *Gadsby et al., 2012*; *Garcia et al., 2012*; *Stanley et al., 2015*), is omitted here, following the approach taken in most previous concentration-dependent models. The layer of complexity that would be added by a voltage dependence of the pump, however, is discussed in the last Results section ('The role of Na+/K+-ATPase voltage dependence').

By measuring sodium currents over time, the pump rates required for ion homeostasis can be estimated (*Joos et al., 2018*, *Equation 19*, Methods). Our model suggests that pump activity sustaining physiological electrocyte firing rates of 200–600 Hz generates a significant hyperpolarizing current, here up to $3\mu A$ (*Figure 1B*). Due to the relatively slow dynamics of ion concentrations, on the timescale of the action-potential generation, this pump current is approximately constant. Therefore, under the assumption of a constant ion channel composition, a strong hyperpolarizing pump current will decrease the input-induced firing rate of the cell, potentially up to the extreme point of silencing it. This effect can be seen in *Figure 1C*, there reflected in a right shift of the frequency-input curve (tuning curve) of a model with electrogenic pump relative to the model without. In other words, for a

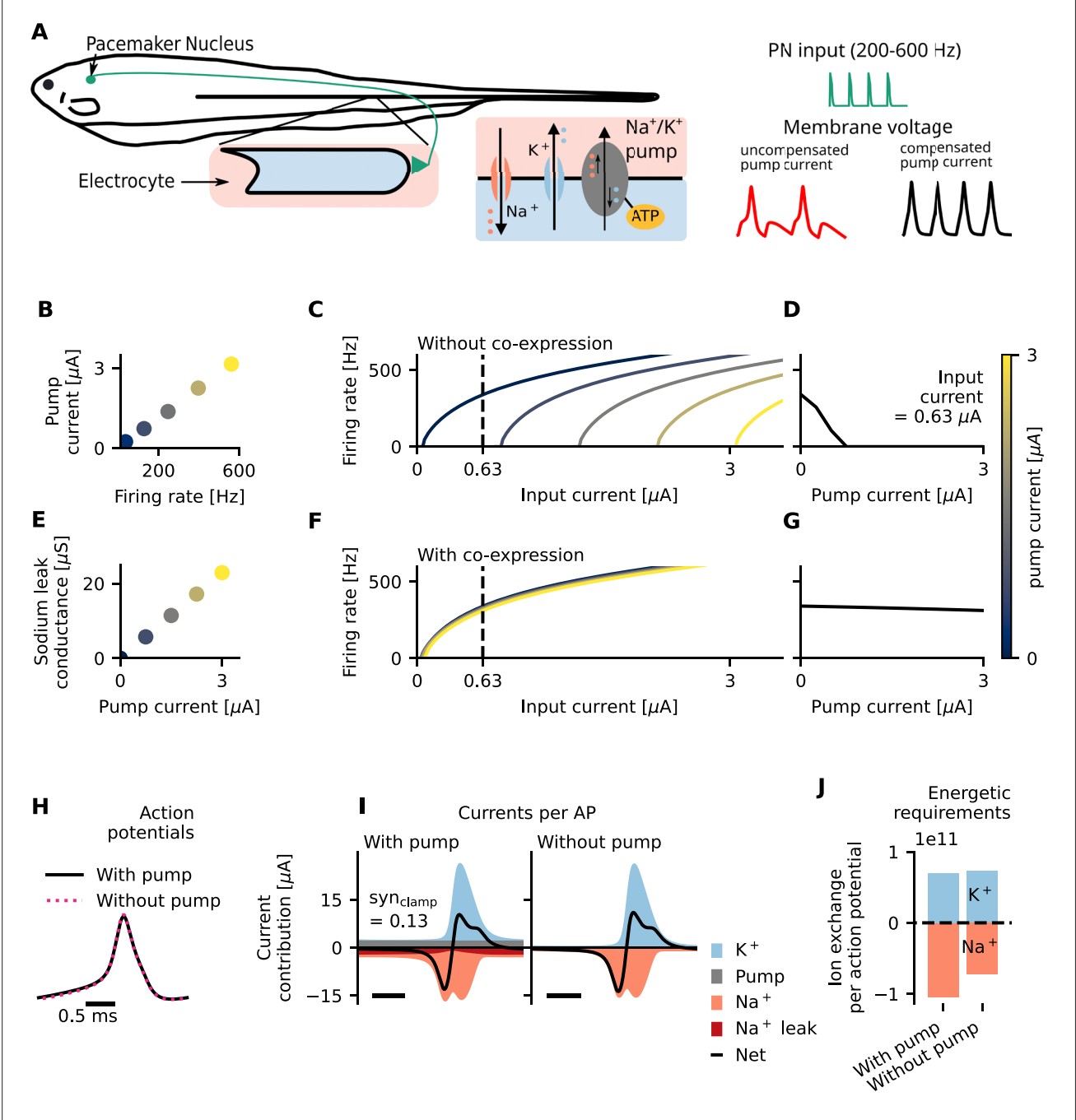

**Figure 1.** In the weakly electric fish electrocyte, Na⁺/K⁺-ATPase electrogenicity requires compensation, which comes at the cost of a more constrained ion channel composition and sub-optimal energetic efficiency. (**A**) The weakly electric fish electrocyte is an excitable cell that locks to high-frequency input from an upstream pacemaker (left). Maintenance of ionic homeostasis is carried out by the Na⁺/K⁺-ATPase (gray), which exchanges three intracellular sodium ions for two extracellular potassium ions and thereby generates a net outward current (center). The compensation of this relatively strong pump current is crucial for faithful synchronization to pacemaker inputs (right). (**B**) High firing rates require significant pump activity, generating a significant hyperpolarizing pump current. (**C**) Increased pump activity, and thus an increased hyperpolarizing pump current, reduces cell excitability because a larger inward input current is needed to activate voltage-gated channels. (**D**) For a fixed physiologically relevant input current (0.63 $\mu$A, see Methods – 'Stimulation in the mean-driven regime') that generates tonic firing in an excitable cell, an increase in pump current `silences' the cell. (**E–G**) Sodium leak channels facilitate a depolarizing current that balances out the hyperpolarizing pump current. If Na⁺/K⁺-ATPase and sodium leak channels are co-expressed (**E**), the impact of increased pump activity on cell excitability is minimized (**F, G**). This is reflected in the impact of the pump current on the tuning curve (**F**) and the impact of the pump current on the firing rate for a fixed, physiologically relevant input current of 0.63 $\mu$A (**G**). (**H–J**) Comparison of action potentials and underlying currents for a constant and physiologically relevant synaptic drive ($\mathrm{syn}_{\mathrm{clamp}}$=0.13, see

*Figure 1 continued on next page*

*Figure 1 continued*

Methods – 'Stimulation in the mean-driven regime') for a model with and without compensated pump current. (**H**) Action potentials are similar in size. (**I**) The additional inward sodium current (dark red) required to balance the outward pump current (gray) results in a simultaneous flow of equally charged ions in opposite directions, decreasing energetic efficiency. (**J**) Effectively, due to this redundancy, more sodium ions per action potential have to be pumped against the gradient.

constant input that elicits high-frequency tonic spiking without this hyperpolarizing pump current, the addition of a pump current decreases a cell's firing rate; for the strong pump currents that occur in highly active cells (*Figure 1B*), firing is eliminated altogether (*Figure 1D*). To maintain high-frequency firing under physiological pump currents (*Figure 1B*), very high inputs are required (*Figure 1C*). These high inputs could come at a significant metabolic cost of synaptic transmission (specifically the cost related to production and packaging of AChR molecules; *Squire et al., 2012*).

Alternatively, the pump-induced raise in current rheobase could be compensated for by adequate adjustments of the cell's ion channel composition. Specifically, for an excitable cell to operate in a regime of tonic firing, the constant net outward pump current could be balanced by an additional constant inward current, which can be achieved via co-expression of Na$^+$/K$^+$-ATPases and, for example, sodium leak channels (*Figure 1E–G*, *Equation 3*, Methods). Although we are not aware of quantitative data on the regulation of ATPase expression in electrocytes, it seems reasonable to assume that the number of pumps expressed in electrocytes scales with the average energetic demand of its spiking activity. An electrocyte that generally fires at higher rates thus requires more pumps to maintain ionic homeostasis. The discharge of the electrical organ (EOD) of *E. virescens* (chosen as a typical representative and whose physiology has been well quantified) approximates to the summed activity of electrocytes. Because individual fish have different baseline EODfs (*Hopkins, 1974*), their electrocyte firing rates also differ. Energetic requirements and, consequently, pump expression levels could therefore also vary among individuals. In turn, pump current and the resulting shift in rheobase (that requires compensation) are likely to be unique for each organism. As can be seen in *Figure 1E–G*, an appropriately chosen co-expression factor between Na$^+$/K$^+$-ATPases and sodium leak channels (*Figure 1E*, *Equation 6*, Methods) suffices to stabilize the rheobase for a wide range of pump currents (*Figure 1F–G*) which are produced within the regime of physiological firing rates (*Figure 1B*). Therefore, such a co-expression mechanism (similar to those that have been described for homeostatic regulation of intrinsic excitability, see for example; *O'Leary et al., 2013*) may provide an elegant solution that allows for reliable tonic high-frequency firing with strong pump activity despite cell-to-cell differences in firing rates and pump expression levels.

The proposed co-expression mechanism not only enables reliable high-frequency firing despite electrogenic pump currents, but also compensates pump currents with minimal effects on AP shape (*Figure 1H*). As the overlap of the relatively constant opposing outward pump current with the compensatory inward current, which comprises one-third of the sum of all inward currents (*Equation 19*, Methods), results in a largely electroneutral exchange of positive ions (*Figure 1I*); however, a surprisingly high fraction of the pump's energy is spent on pumping out sodium ions that do not directly contribute to the action potential of the cell but only compensate for the additional pump current. Therefore, the energetic efficiency of action potentials is reduced (*Yu et al., 2012*) because of the electrogenicity of the Na$^+$/K$^+$-ATPase by one third compared to the hypothetical scenario of electroneutral pumping, where no additional inward currents would be needed to enable tonic firing (*Figure 1J*). In absolute terms, this effect is particularly severe for systems operating at high average firing rates which require high pump densities to maintain ionic homeostasis (*Figure 1B*) and even higher pump densities to additionally maintain spike amplitudes (here of around 13 mV; *Joos et al., 2018*; Appendix 1).

In a tonically active cell, the negative feedback loop that the electrogenic Na$^+$/K$^+$-ATPase provides to enhance ionic homeostasis for action-potential firing could thus come at the cost of a more constrained ion channel composition and sub-optimal energetic efficiency (see Appendix 4 for further discussion on metabolic costs).

## Na$^+$/K$^+$-ATPase affects the tuning curve

As outlined above, quantitatively, the compensation required because of the pump's electrogenicity depends on a cell's spiking activity. Consequently, even if ion channels and Na$^+$/K$^+$-ATPases were

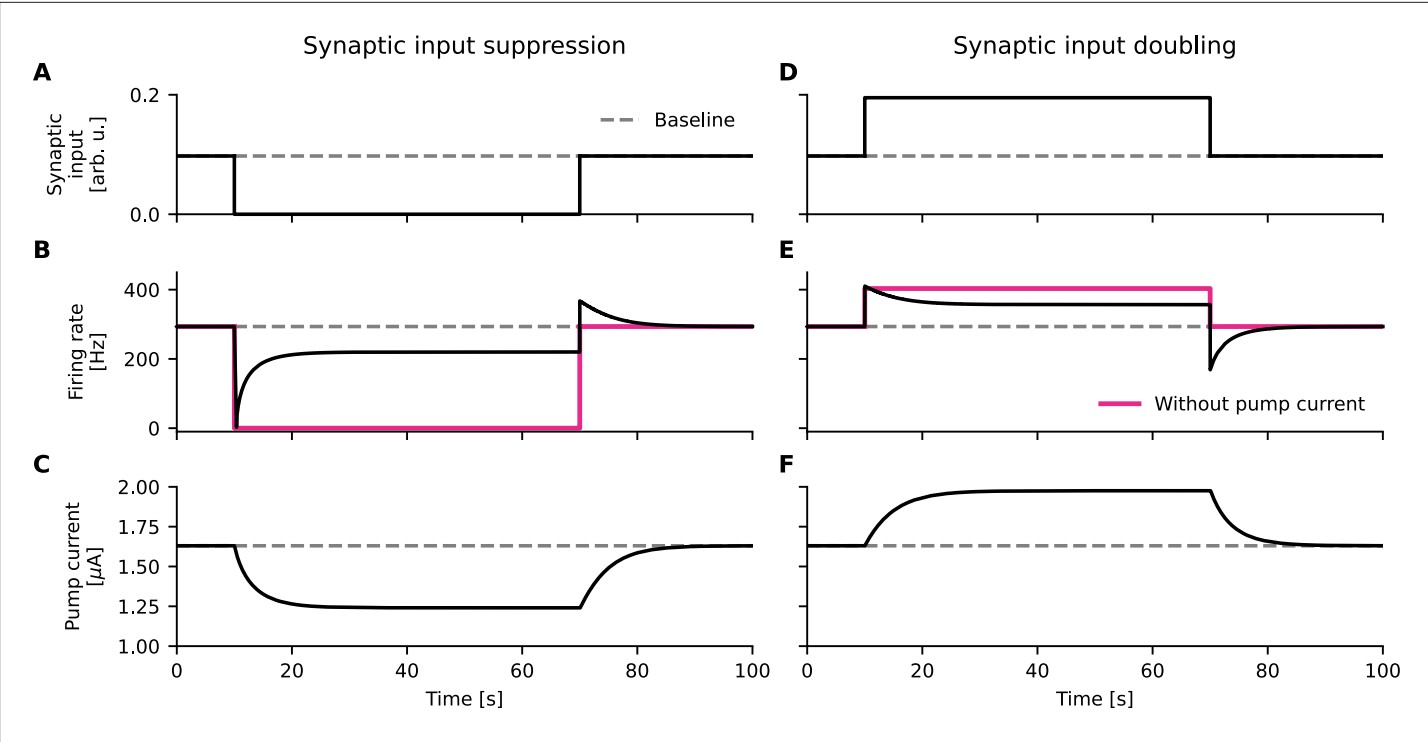

**Figure 2.** Homeostatic feedback loops based on Na+/K+-ATPase activity affect firing responses through altered pump currents. (**A, B, C**) Synaptic input suppression (**A**) initially silences the cell (**B**). The Na+/K+-ATPase adjusts to the reduction in energetic demand through reduced activity. This reduces the pump current (**C**), which increases cell excitability and results in spontaneous firing without synaptic inputs (B, black). Without a pump current (magenta), spontaneous firing is not induced. (**D, E, F**) Increased synaptic inputs (**D**) initially increase firing rates (**E**). The Na+/K+-ATPase adjusts to the increase in energetic demand through increased activity. This increases the pump current (**F**), which decreases cell excitability and results in reduced firing rates (E, black).

co-expressed (*Equation 6*, Methods), and both were optimized to facilitate tonic firing in an excitable cell (*Equation 19*, Methods), the electrogenicity of the Na+/K+-ATPase still could interfere with neural processing when firing rates change drastically due to input changes, as outlined in the following.

Most excitable cells do not operate at a fixed firing rate. Often, a flexible, transient modulation of firing rates is required either to track and encode stimuli (*Hildebrandt et al., 2011*) or to control and adapt motor programs (*Hürkey et al., 2023*). Such a modulation is evoked by alterations in synaptic inputs that differ from baseline activity (*Figure 2A and D*). If the pump rate remains constant despite such changes in firing rates, ion accumulation or drain is to be expected. This can lead to critical transitions in cell excitability and function (as intrinsic cell dynamics depend on ion concentrations; *Contreras et al., 2021*; *Behbood et al., 2023*; *Barreto and Cressman, 2011*; *Kager et al., 2000*), and, in the extreme case, diminish the ion concentration gradient to the extent that firing is completely impaired (*Donnan, 1924*).

At first glance, the pump's sensitivity to ionic concentrations (*Equation 20*, Methods) seems an adequate solution that can alleviate such effects of drastically changing firing rates. The dependence of the pump on ionic concentrations contributes to an activity-dependent restoration of ion gradients, that is, an appropriately calibrated concentration dependence of Na+/K+-ATPase activity can help to match the energetic demand of the cell's recent activity (*Crambert et al., 2000*; *Figure 2C and F*). The adapted pump activity, however, is accompanied by a change in hyperpolarizing current; the cell is pushed to a regime that the system was not originally tuned to (*Figure 2B and E*). Therefore, even in a perfectly controlled environment, an excitable cell could assume different firing rates in response to the same input, where the immediate output of the cell depends on previous activity (*Megwa et al., 2023*). If a fixed input-output mapping is key to the function of an excitable cell, the electrogenicity of the Na+/K+-ATPase may induce yet another trade-off between ionic homeostasis and cell function.

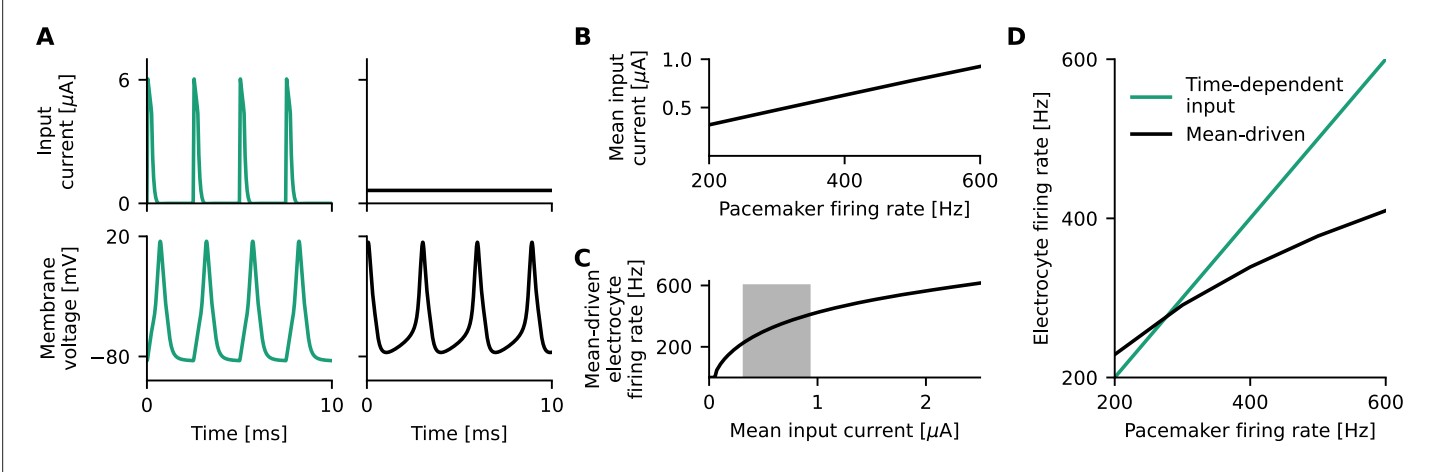

**Figure 3.** The electrocyte operates in a mean-driven regime, and its mean-driven properties affect its entrainment to periodic inputs from the Pacemaker Nucleus (PN). (**A**) The input current to the electrocyte stemming from the PN (top, left) sets the high-frequency firing rate of the electrocyte (bottom, left). Constant input currents (top, right) also elicit tonic high-frequency firing (bottom, right). (**B**) Mean input currents stemming from physiologically relevant PN inputs of 200–600 Hz. (**C**) High-frequency electrocyte firing is realized for constant input currents that lie within the mean of the input currents that are generated in the behaviorally relevant regime (200–600 Hz) (gray box, **B**). (**D**) There is a frequency mismatch between the pacemaker firing rate and the mean-driven electrocyte firing rates, which influences signal entrainment.

## Na⁺/K⁺-ATPase affects entrainment

As we illustrate next, a pump-induced alteration of response properties of excitable cells could be especially problematic if these cells are to be entrained in a network or by a pacemaking system. Weakly electric fish electrocytes, like most excitable cells, do not operate in isolation. In order to create an oscillating weakly electric field, electrocyte firing needs to be coordinated across their population, which is enabled by a common drive from an upstream pacemaker. In order to serve a variety of communication paradigms with largely different EOD patterns and hence electrocyte firing rates, an accurate manipulation of the electrocyte firing rates by the pacemaker nucleus is crucial for electric fish.

Each electrocyte in the electric organ is innervated on the posterior side by the spinal motor nerve, which transmits signals from the pacemaker nucleus to the electrocyte (*Figure 1A*; *Ban et al., 2015*). Electrocytes are not synaptically connected among each other; they receive a unidirectional synaptic input from the pacemaker nucleus and firing patterns are only driven by the pacemaker. Therefore, we can model the effects of upstream cells on the electrocytes by simulating the periodic input currents that originate from neurotransmitter release in the synaptic cleft, modulated by pacemaker nucleus activity (*Figure 3A*, top left, *Equation 26*, Methods; *Joos et al., 2018*). The pacemaker entrains the electrocyte on a spike-by-spike basis, and the electrocyte firing rates should faithfully follow the firing rates of the pacemaker (*Figure 3A*, bottom left) to give rise to a strong, high-amplitude EOD signal.

The model suggests that the electrocyte operates in a mean-driven regime, that is, the mean of the time-varying input it receives from the pacemaker suffices to invoke tonic spiking in the electrocyte (*Figure 3A–C*). Whether the electrocyte is entrained by the pacemaker depends on characteristics of the electrocyte's voltage dynamics (like the susceptibility to perturbations, reflected in the so-called phase-response curve) as well as the frequency mismatch between the pacemaker and the firing rate of the electrocyte in response to the stimulus mean. If the frequency mismatch is too large, entrainment fails (*Figure 3D*, *Appendix 2—figure 1*). As the pump current affects the mean-driven firing rates of the electrocyte (*Figure 2*), it can significantly impact entrainment in this simple network, in particular, when the pacemaker frequency and hence mean input to the electrocyte changes.

For an individual *E. virescens*, pacemaker firing rates can remain constant over long periods of time (*Bullock, 1970*; *Moortgat et al., 2000*). If the electrocyte repeatedly receives the same input and thus produces action potentials that displace a fixed amount of ions per unit of time, pump rates and co-expressed inward leak channels could be tuned to maintain tonic firing and ionic homeostasis (*Equations 6; 19*, Methods). When searching for food, hiding from predators, and courting, however,

substantial deviations from baseline occur. In such cases of drastic change in pacemaker firing rate, the pump rate and thus the pump current adapts via its concentration dependence to the new firing statistics of the electrocyte which, consequently, alters the tuning curve (*Figure 1C*) and hence also the mean-driven firing rates (*Figure 2*). In electrocytes of *E. virescens*, behaviorally relevant deviations from baseline firing come in several forms which include chirps and frequency rises (both used for inter-individual communication) and have different consequences for cell entrainment, as we illustrate in more detail in the following.

## The Na$^+$/K$^+$-ATPase affects the ability to generate EOD chirps

*E. virescens* chirps consist of short cessations (type A) or period doublings (type B) of the EOD and are thought to play an important role in dominance fights and courtship (*Hopkins, 1974*; *Hagedorn and Heiligenberg, 1985*; *Stöckl et al., 2014*). Type A chirps, which essentially correspond to short 'pauses' in EOD generation, are generated in electrocytes through short interruptions of PN firing (*Figure 4A*, see Methods - Periodic stimulation; *Szabo and Enger, 1964*). As electrocytes are only innervated by excitatory synapses, successful chirp generation thus relies on an electrocyte that is 'silent' when devoid of input.

From experimental observations, it is known that the length of type A chirps in *E. virescens* can extend beyond 20 times the length of one EOD (*Hopkins, 1974*; *Stöckl et al., 2014*). Repetitive emission of such long type A chirps (*Figure 4B*, top) decreases mean firing rates of electrocytes and thereby the action-potential-induced ion displacement, ultimately resulting in a lowered pump current (*Figure 4B*, bottom). We find that in our model, the effect of an individual chirp on pump currents is small and does not alter electrocyte excitability to an extent that firing rates are severely affected (*Figure 4C*, left). In case of consecutive chirping, however, the hyperpolarizing pump current progressively weakens with time (*Figure 4B*, bottom), eventually leading to spontaneous electrocyte firing in absence of pacemaker input (*Figure 4C*, right). This effect limits the number and duration of chirps that can be induced by the pacemaker (*Figure 4B*, top). The observed dynamics suggest that mechanisms increasing the timescale of the pump feedback loop, such as extracellular potassium buffering (*Equation 21*, Methods), are suited to diminish the effects of chirps on the pump current (*Figure 4D*) and thereby the effect of variable input signals on chirp generation (*Figure 4E*). We find that extracellular potassium buffering is particularly efficient in dampening Na$^+$/K$^+$-ATPase effects on cell excitability, because Na$^+$/K$^+$-ATPase rates of excitable cells are especially sensitive to extracellular potassium concentrations (*Equation 20*, Methods; *Erecińska and Dagani, 1990*; *Hübel and Dahlem, 2014*) and potassium buffering reduces the variability of potassium concentrations in extracellular space. Metabolic costs of potassium buffering, however, may constitute an additional expense in the total energy budget of the organism (see Appendix 4 for further discussion).

## The Na$^+$/K$^+$-ATPase affects generation of frequency rises

During courtship behavior, sudden frequency rises of the EOD followed by an exponential frequency decay back to baseline in the course of 2–40 s are thought to constitute an important signal (*Hagedorn and Heiligenberg, 1985*). Frequency rises are produced by the pacemaker, and the increase in PN firing rate is meant to entrain the electrocyte accordingly (*Figure 5A*). To this end, the mean-driven firing rates of the electrocyte should be sufficiently similar to the transiently elevated PN firing rates, because entrainment fails if frequency mismatches are too large (*Figure 3D*, *Pikovsky et al., 2002*; *Schreiber et al., 2004*). Accordingly, electrocytes with very slow mean-driven dynamics cannot be entrained to very fast PN inputs (see *Appendix 2—figure 1*).

Short frequency rises in *E. virescens* of around 2 s have been measured to encompass frequency elevations of up to 40 Hz (*Hopkins, 1974*). Repetitive emission of such frequency rises (*Figure 5B*, top) increases mean firing rates and thereby the action-potential-induced ion displacement, resulting in an increased pump current (*Figure 5B*, bottom). Comparable to the observation for chirps, we find that in our model, the effect of a single frequency rise on pump currents is small and does not alter electrocyte excitability to an extent that impedes entrainment (*Figure 5C*). With repetition of these communication signals, however, the hyperpolarizing pump current significantly increases over time (*Figure 5B*, bottom), eventually decreasing the electrocyte's mean-driven firing rate to the point where a precise 1:1 locking between PN and electrocyte breaks down (*Figure 5D*, entrainment index is statistically smaller than in C; *Fisher, 1993*). Again, this effect of the pump imposes a limit on the number

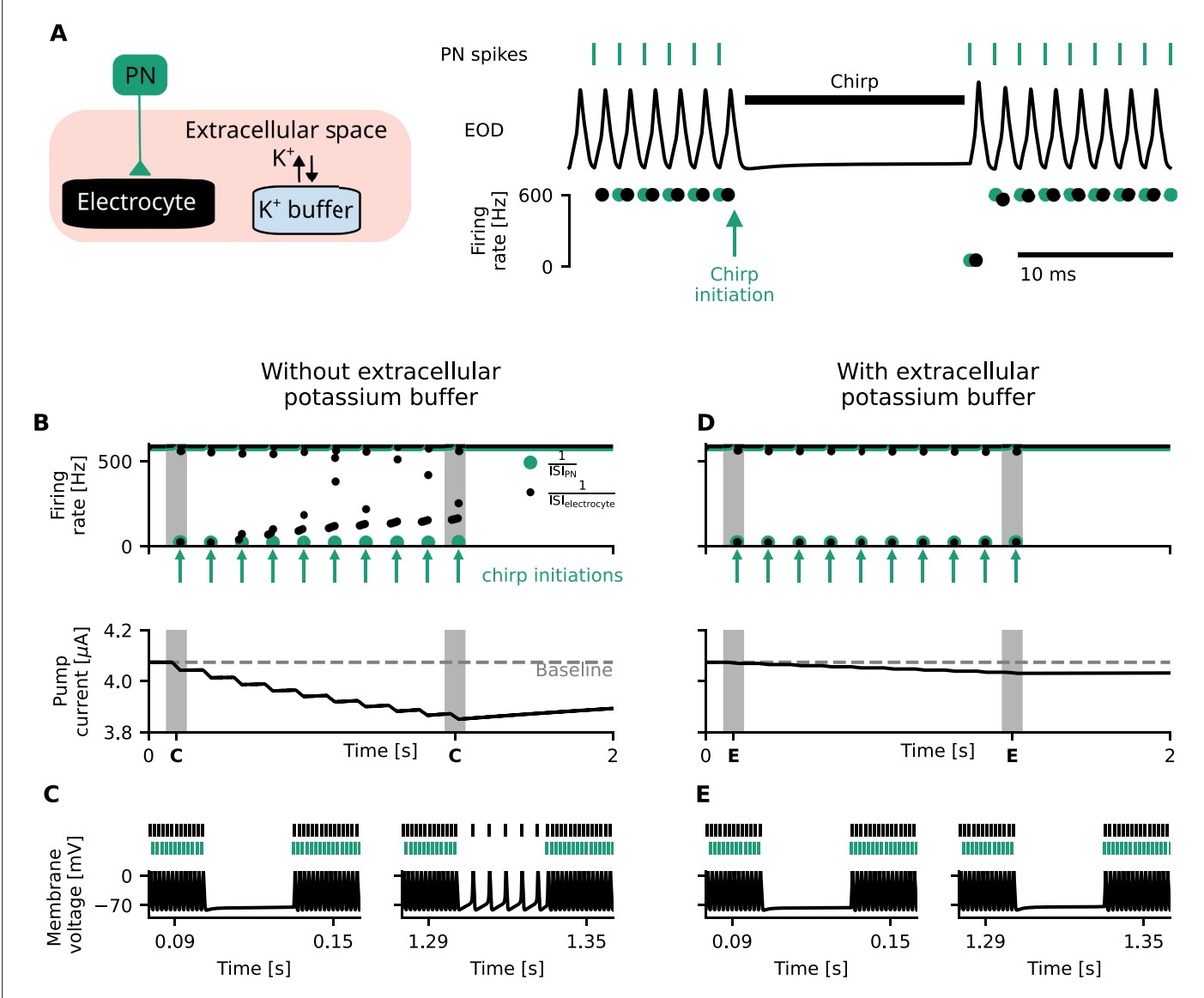

**Figure 4.** Homeostatic feedback loops on Na⁺/K⁺-ATPase activity impede chirp generation in electrocytes and can be mitigated through extracellular potassium buffering. (**A**) Schematic illustration of the chirp setting. Left: the electrocyte (black) is coupled to the pacemaker nucleus (PN, green) with an excitatory synapse. A potassium buffer (blue) regulates extracellular potassium concentrations. Right: PN spikes (green, top) induce chirps in the electrocytes through cessation of inputs and thereby temporarily shut off the Electric Organ Discharge (EOD, middle). When chirps are properly generated, instantaneous firing rates (bottom) of the electrocyte (black) equal those of the PN (green). (**B**) The pacemaker generates 10 consecutive chirps, indicated by green arrows and instantaneous PN firing rates $\frac{1}{\overline{ISI}_{PN}}$. This lowers the mean firing rate of the electrocyte (black, top) and thereby its energetic demand. Consequently, the pump current decreases over time (bottom). This decreased pump current increases cell excitability, which over time (in this paradigm after 400 ms) leads to a mismatch between PN and electrocyte firing rates (top). (**C**) Electrocyte (black) and PN (green) spikes (top) and electrocyte membrane voltage (bottom) during chirps before (left) and after (right) a significant decrease in the excitability-altering pump current. After such a deviation in pump current, electrocyte firing occurs during chirps (right). (**D, E**) Same as (**B, C**) with extracellular potassium buffering. Extracellular potassium buffering extends the timescale of the homeostatic feedback loop of Na⁺/K⁺-ATPase activity on energetic demand, which reduces the effect of transient firing-rate deviations on the pump current (**D**, bottom). This minimizes the deviation in pump current to the extent that chirps can be reliably generated (**D**, top), (**E**).

and duration of such frequency rises that can be induced without impairment of electrocyte entrainment and hence the EOD strength (*Figure 5B*, top). Mechanistically, the ability of the electrocyte to entrain to the pacemaker does not only depend on their frequency mismatch, but also on the strength of their synaptic coupling (Appendix 2, *Equation 33*). An increase in synaptic coupling strength (see

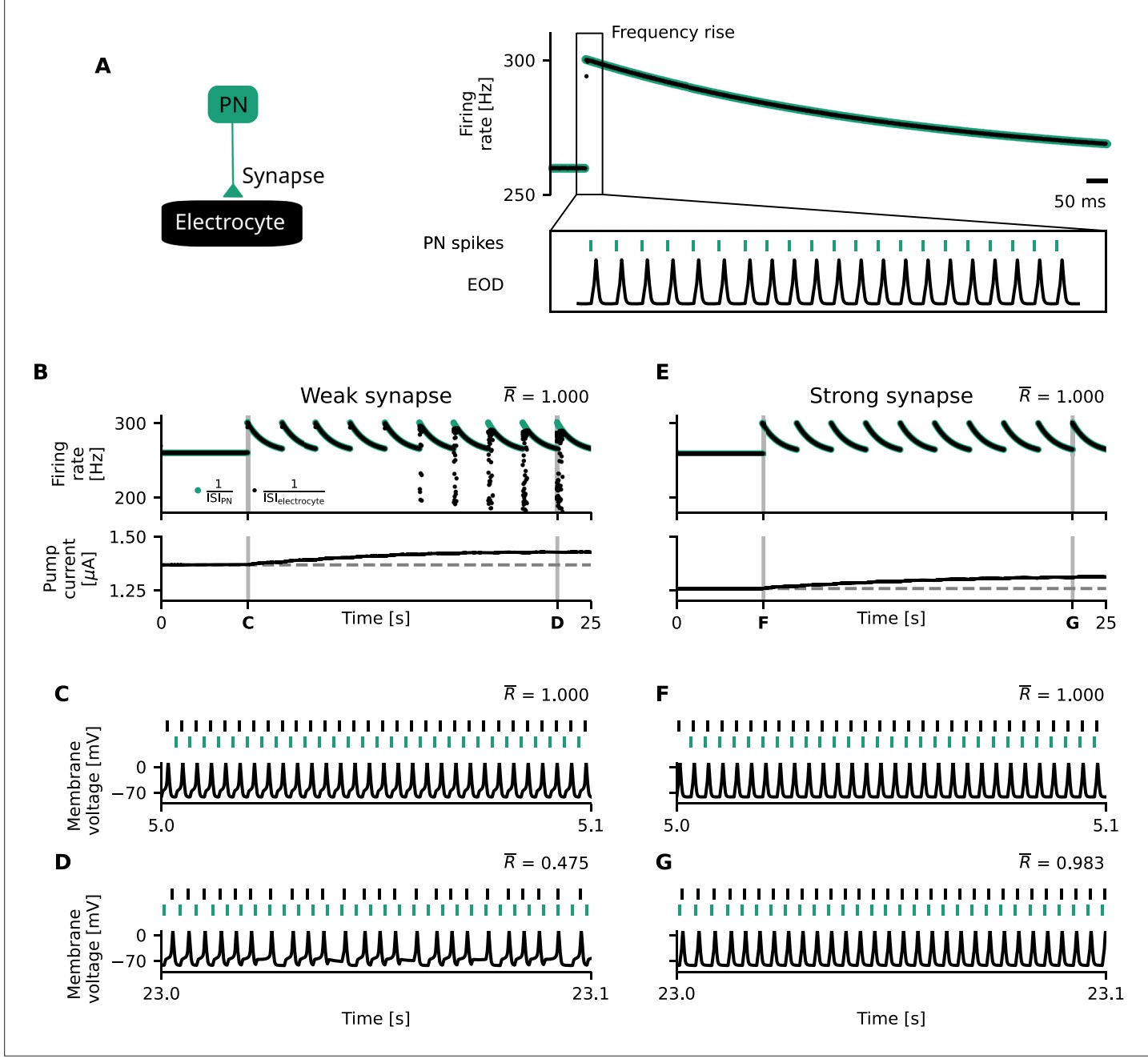

**Figure 5.** Homeostatic feedback loops on Na$^+$/K$^+$-ATPase activity impede the generation of frequency rises and can be mitigated through strong synaptic coupling. (**A**) Schematic illustration of the generation of frequency rises. Left: the electrocyte (black) is coupled to the pacemaker nucleus (PN, green) with an excitatory synapse. Right: Frequency rises are generated through a rapid increase in PN firing rates which exponentially decay back to baseline rates (green, top). As the electrocytes are entrained by the PN (bottom), their firing rates mimic that of the PN and also show a frequency rise (black, top). (**B**) The generation of consecutive frequency rises by the pacemaker (green) increases the mean firing rate of the electrocyte (black, top) and thereby the energetic demand of the electrocyte, which is fed back into an increased pump current (bottom). This increased pump current decreases cell excitability, which over time (in this paradigm after 15 s) leads to a mismatch between PN and electrocyte firing rates during the frequency rises (top). Overall, however, synchronization is very stable, which is reflected in the synchronization index $\overline{R}$ (*Equation 29*, Methods). (**C, D**) Electrocyte (black) and PN (green) spikes (top) and electrocyte membrane voltage (bottom) during frequency rises before (**C**) and after (**D**) a significant increase in excitability-altering pump current. After a significant deviation in pump current, not all PN spikes are reproduced in the electrocyte which leads to 'missing' spikes (**D**). This is reflected in the synchronization index $\overline{R}$, which decreases with increasing pump current deviation. (**E–G**) Same as (**B–D**) with strong synaptic coupling. Strong synaptic coupling attenuates the effect of altered pump currents on electrocyte entrainment and enables reliable production of frequency rises (**E**, top), (**F, G**).

Methods - Mechanisms that improve entrainment) extends the maximum frequency mismatch that still allows for synchronization (Appendix 2, , *Equations 37; 38*). Both effects of frequency mismatch and synaptic coupling can be illustrated by the so-called Arnold tongue (*Pikovsky et al., 2002*). We therefore hypothesize that a strong synapse facilitates electrocyte entrainment and could prolong phases without pump-induced entrainment breakdown (*Figure 5E–G*). Yet, it comes at the energetic cost of the increase in neurotransmitter release (including the production and packaging of AChR molecules, see Appendix 4for further discussion; *Squire et al., 2012*).

The analysis of both types of signals, chirps and frequency rises, shows that the electrogenic Na$^+$/K$^+$-ATPase can have significant effects on the computational properties of highly active excitable cells, potentially requiring energetically costly countermeasures for normal operation, especially if cells are to be entrained by a pacemaker or in a network. This suggests that even though Na$^+$/K$^+$-ATPase was jury-rigged to support the generation of action potentials in excitable cells, the fact that their original function required them to be electrogenic inevitably calls for countermeasures that lower the energetic efficiency of signaling.

## The role of Na$^+$/K$^+$-ATPase voltage dependence

A pump current that only varies on the longer time scale of changes in ion concentrations acts like a constant current on the time scale of spike generation (*Figure 1I*) and will horizontally shift the tuning curves as described above (*Figure 1C*). The consequence of this shift (if uncompensated as described above) is a drastic change in firing rates (*Figures 1D and 2*, see Methods - Modeling the pump current and sodium leak channel co-expression). We next explore whether changes of pump rates on the shorter timescale of action potentials can alleviate the pump-current induced firing-rate adaptation. In particular, we illustrate in the following that a voltage dependence of the pump may constitute an interesting means to limit pump-induced firing-rate modifications and at the same time save on the energetic cost of action potentials.

A common supposition is that (to restore the ionic gradients that get depleted during the generation of action potentials) the pump only depends on ion concentrations and hence, to first approximation, displays constant activity during spiking. In this case, the hyperpolarizing pump current counteracts the sodium currents at the depolarization phase of the AP upstroke and, in fact, assists the potassium currents in repolarizing the cell during the action-potential downstroke (*Figure 6A*, left). In the following, we contrast this constant pump to a voltage-dependent pump that activates selectively only during the action-potential downstroke (*Figure 6A*, right). The pump's voltage dependence could benefit a neuron in two ways: First, by not affecting sodium-based depolarization, it may reduce the shift that the pump current induces in the tuning curve. Second, by aiding potassium in repolarization, it could provide some energetic benefits for spiking cells.

The exact kinetics and voltage dependence of Na$^+$/K$^+$-ATPases differ per cell type and organism (*Garcia et al., 2012*; *Stanley et al., 2015*; *Colina et al., 2010*) and have likely evolved differently in distinct cells to support their unique function and energetic demand. To highlight the positive effects of a voltage dependence of the Na$^+$/K$^+$-ATPase in electrocytes, whose dependence on voltage to our knowledge currently is unknown, we use a thought experiment based on an idealized voltage-dependent pump that optimally compensates tuning curve shifts and reduces energy demand (*Figure 6A*, right).

Specifically, the dynamics of this idealized voltage-dependent pump is assumed to exactly mimic the hyperpolarizing potassium current in the following way: Start with a classical Hodgkin and Huxley equation without pump. Then reduce the repolarizing potassium current to $\frac{2}{3}$ of its strength. The missing $\frac{1}{3}$ is now substituted by a pump current with exactly the same time course $I_\mathrm{p} = \frac{1}{3}I_\mathrm{K}$, such that,

$$C_\mathrm{m}\dot{v} = -I_\mathrm{Na} - I_\mathrm{K} = -I_\mathrm{Na} - \frac{2}{3}I_\mathrm{K} - \frac{1}{3}I_\mathrm{K} = -I_\mathrm{Na} - \frac{2}{3}I_\mathrm{K} - I_\mathrm{p}.$$

In the model without pump, the cumulative sodium and potassium currents have to add up to zero after one period (Appendix 3). This, together with the equation above, implies that the chosen $I_\mathrm{p}$ meets the requirements of the pump stoichiometry $\oint I_\mathrm{p}dt = \frac{1}{3} \oint I_\mathrm{Na}dt$ and $\oint I_\mathrm{p}dt = \frac{1}{2} \oint I_\mathrm{K}dt$ and thereby perfectly counteracts currents flowing during a complete action-potential cycle to maintain ion homeostasis. The equation also shows that replacing one-third of the potassium channels with the idealized voltage-dependent Na$^+$/K$^+$-ATPase would leave the action potential shape unchanged

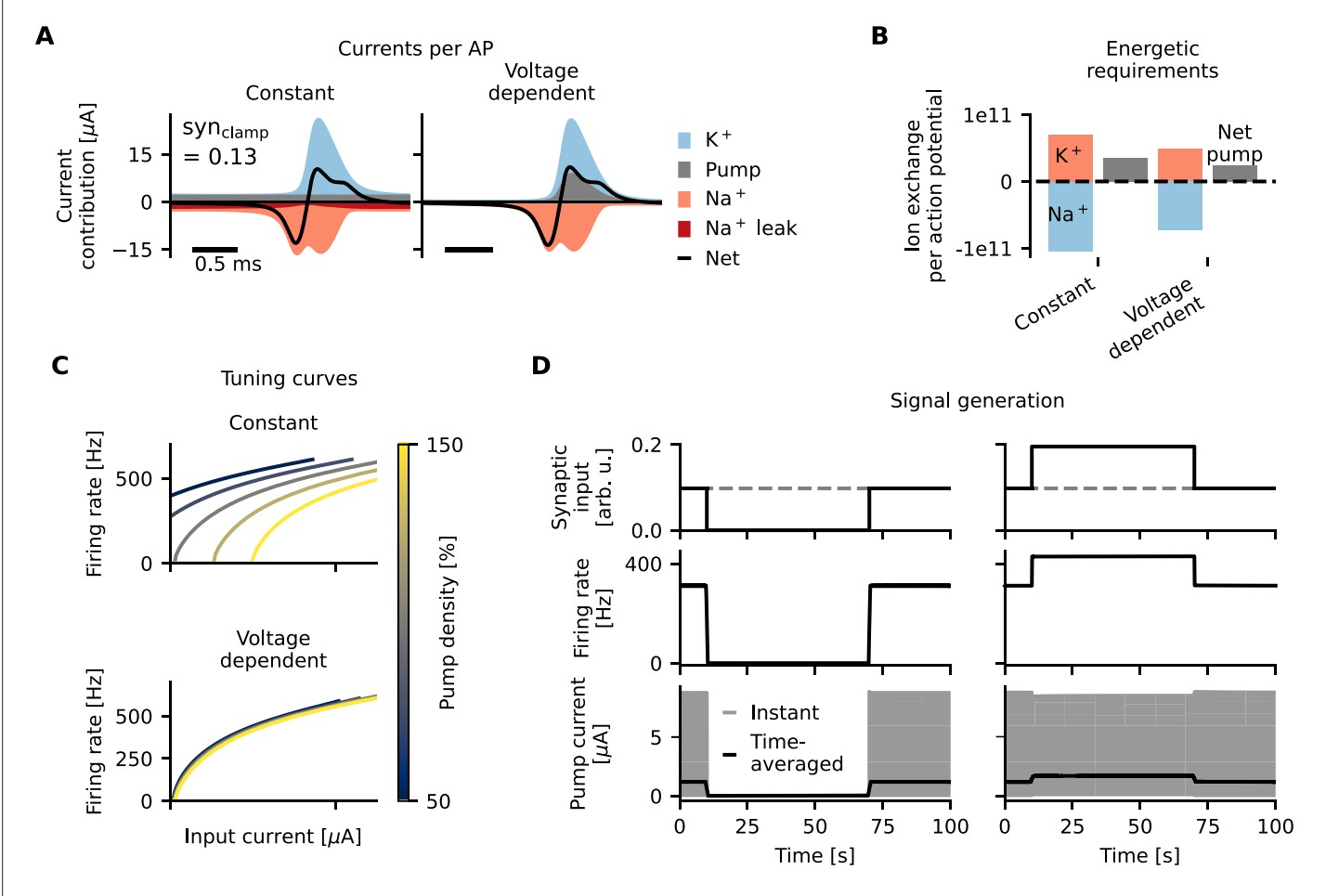

**Figure 6.** Ideal voltage dependence of the Na$^+$/K$^+$-ATPase for energy-efficient action potentials and minimal firing-rate adaptation. (**A**) Action potential current contributions to the total in- and outward currents for a constant and physiologically relevant synaptic drive (syn$_{clamp}$=0.13, see Methods - Stimulation in the mean-driven regime) with a Na$^+$/K$^+$-ATPase without voltage dependence (left) and with an optimal voltage dependence that mimics potassium channels (right). The voltage-dependent pump current takes on the role of a potassium channel and contributes significantly to the net current at the AP downstroke. (**B**) Total amount of sodium (red) and potassium (blue) ions, and net ion transfer of the pump (gray) that are relocated per AP for a cell with a voltage-insensitive pump (left) and a voltage-dependent pump (right). (**C**) The effect of pump density on the tuning curve is minimal for ideal voltage-dependent pumps (bottom) compared to a non-voltage-dependent pump (top). (**D**) Signal generation in a cell with Na$^+$/K$^+$-ATPases with optimal voltage dependence. Synaptic input suppression (top left) silences the cell (center left) and reduces the pump current (bottom left). Firing rates are, however, not changed, and the cell remains silent. Synaptic input doubling (top right) increases firing rates (center right) and increases time-averaged pump currents (bottom right). Firing rates are, however, not affected. Note that the instantaneous pump current (bottom, gray) varies on the timescale of action potentials, which is highly compressed in this 100 s time window.

compared to the model without pump. Importantly, as the pump substitutes for potassium channels, it reduces the flux through these channels by a factor of $\frac{1}{3}$. In addition, compared to the constant pump scenario, there is no need for sodium leak channels to cancel out the hyperpolarizing pump current. This additionally reduces the cumulative flow of sodium ions by approximately $\frac{1}{3}$ with respect to an excitable cell with relatively constant pumping. Taken together, the reduction in flow of sodium and potassium ions reduces the pump load by approximately $\frac{1}{3}$ (*Figure 6B*).

Besides lowering the energetic demand, voltage sensitivity can also reduce the effect of the Na$^+$/K$^+$-ATPases on the neuron's tuning curve and, consequently, firing-rate adaptation. Two main factors help reduce the adaptation: First, the pump current with a potassium-like voltage dependence is almost completely inactive at spiking onset (approximately –80 mV), in contrast to the fully active constant pump current. Alterations in the former pump current hence induce almost no shift in the tuning curve (*Figure 6C*). Second, as the pump optimally scales with membrane currents at any firing rate in the voltage-dependent case, it instantaneously leads to near-perfect homeostasis. Therefore,

action-potential-induced concentration changes are minimal. In contrast to the case of a constant pump (**Figure 2**), pacemaker-induced jumps in electrocyte drive hence do not elicit a substantial firing-rate adaptation because concentrations remain largely unchanged (**Figure 6D**). We note that even if they would, their effect would be small due to the first argument (**Figure 6C**).

Consequently, suppression of synaptic inputs (like during a chirp) does not result in spontaneous firing for a pump with the described voltage dependence and a cell can be reliably 'silenced' for extended periods of time (**Figure 6D**, left). Furthermore, there is a fixed input-output mapping from synaptic input to firing rates in this case (**Figure 6D**, right), which benefits the robustness of entrainment between pacemaker and electrocyte.

While it remains to be established experimentally whether $Na^+/K^+$-ATPases in electrocytes do exhibit a voltage dependence resembling the one postulated here, our thought experiment demonstrates the generic potential of a voltage dependence of pumps to mitigate negative side effects of electrogenic pumping in highly active cells and to lower the need for a costly investment into alternative compensatory measures such as the co-expression of additional ion channels, increased synaptic weights, or extracellular potassium buffering. A voltage dependence of the hypothesized ideal pump, however, does require a higher pump density. As the idealized pump's activity is constrained to specific time windows, its peak pump rate during these periods needs to be higher. In our model, the effective pump rate during these (limited) times is elevated by a factor of four in comparison to a voltage-agnostic yet constantly active pump (**Figure 6A**). This brings other, potentially significant constraints, such as available membrane space and the cost of pump synthesization and transport to the table (see Appendix 4 for further discussion).

## Discussion

We investigated the rarely acknowledged side effects of the electrogenic $Na^+/K^+$-ATPase on the computational properties of a highly active spiking cell: the weakly-electric fish electrocyte. Our findings highlight that the electrogenicity of the $Na^+/K^+$-ATPase may pose challenges for robust signal processing in highly active cells; for such cells, a pump that would have evolved for the sole purpose of maintaining ionic gradients would be more efficient if it was electroneutral. We dissect the mechanisms involved and show that energy-intensive countermeasures may be required to ensure robust performance in the presence of the pump when a tonically active cell is driven by inputs of alternating mean: a perspective that is underrepresented in the current literature. Specifically, for the weakly electric fish electrocyte, where robust performance is crucial for intraspecific communication, we formulate the following testable hypotheses: (i) $Na^+/K^+$-ATPase is co-expressed with ion channels that facilitate a relatively constant inward current (such as sodium leak channels), (ii) in this organism, $Na^+/K^+$-ATPase dynamics have evolved to act in a voltage-dependent manner on the time scale of action potential firing, and (iii) extracellular potassium is buffered, or $Na^+/K^+$-ATPase sensitivity to this ion is reduced in comparison to that in other organisms. While analyzed in electrocytes, the model is and identified mechanisms are sufficiently generic to translate to other excitable cells across the animal kingdom operating at high firing rates, such as Purkinje neurons (**Person and Raman, 2012**), vestibular nuclear neurons (**Gittis et al., 2010**), and fast-spiking interneurons (**Roth and Hu, 2020**; **Anderson et al., 2010**).

When Hodgkin and Huxley established their pioneering conductance-based model of action potential generation in 1952 (**Hodgkin et al., 1952**), it was generally assumed that the sodium-potassium pump, which maintains the ionic gradient across the cell membrane, was electroneutral (**Thomas, 1972**). Their computational model, which was adapted over the following 70 years to model numerous types of excitable cells in diverse tissues and species, therefore did not include a pump current. Even later, after the 3:2 stoichiometry of the sodium-potassium pump and thus its electrogenicity were proven, the pump current was often not included in simple point-neuron models, presumably because of its relatively small amplitude and effect size (**Hodgkin and Keynes, 1955**; **Hodgkin and Keynes, 1956**). Although this may be a suitable argument for excitable cells that are only moderately active, it is less likely to hold for cells that need to be tonically active over long periods of time. In fact, tonically active cells are expected to operate closer to the strong-activity-inducing, post-tetanic stimulation protocols used in the 1960s to render the pump current more visible by artificially increasing their effect size (**Straub, 1961**; **Nakajima and Takahashi, 1966**; **Gage and Hubbard, 1966**). Additionally, the functional consequences of other, more subtle pump properties, such as its voltage dependence,

have not been explored. Here, we identify a range of possible compensation mechanisms for strong pump currents and speculate that, at least theoretically, the pump itself may hold a key to rendering its effects less computationally invasive.

## Generalization to other cell types

In this article, two statements were made on the electrogenic pump: in highly active cells, its hyperpolarizing current can be strong enough to interfere with signal coordination, and several biophysical cell properties can be exploited to diminish this effect. Assuming that the Na$^+$/K$^+$-ATPase carries out the majority of active sodium- and potassium transport, on average, the pump current is roughly a third of the sum of all sodium currents (*Equation 19*, Methods). This holds for any excitable cell (under the above-mentioned assumption), regardless of their ion channel composition, channel biophysics, and additional pumps and transporters. The magnitude of the sum of sodium currents, and thus of the pump current, however, not only depends on a cell's firing rate, but also on the dynamics of all currents that contribute to the action potential. The time separation between inward sodium and outward potassium currents, for example, dictates AP energy efficiency (*Yu et al., 2012*) and thus the pump load. Therefore, excitable cells with different sodium channel dynamics than shown here, such as Purkinje cells with resurgent sodium (*Khaliq et al., 2003*), are likely to exhibit a different degree of crosstalk between the pump current and AP-generating currents, depending on the resulting redundancy in sodium and potassium flows. Moreover, although other ions such as calcium and chloride are assumed not to play a significant role in electrocyte action potentials (*Ferrari and Zakon, 1993*; *Dunlap et al., 2017*), other excitable cells such as mammalian neurons have been shown to be influenced by these ions (*Bean, 2007*; *Huang et al., 2012*).

The main players in calcium removal, which is required to maintain calcium gradients, are the Ca$^{2+}$-ATPase and the Na$^+$/Ca$^{2+}$ exchanger (*Strehler and Zacharias, 2001*). Due to its 3:1 stoichiometry, the combined activity of the Na$^+$/Ca$^{2+}$ exchanger and the Na$^+$/K$^+$-ATPase effectively results in an electroneutral active transport of calcium ions. Calcium that is transported through the Ca$^{2+}$-ATPase, however, comes with the by-product of a strong hyperpolarizing current that will induce the same effects on spiking behavior as the Na$^+$/K$^+$-ATPase currents presented in this article. Therefore, in cells where calcium currents contribute significantly to its AP generation, the ratio of Ca$^{2+}$-ATPase to Na$^+$/Ca$^{2+}$ exchanger expression might be crucial for cell function. The quantitative influence of other ions such as calcium and chloride is, however, to be determined in future work.

In general, we can state that as long as sodium ions are the main player in a cell's AP generation, and if the Na$^+$/K$^+$-ATPase carries out the majority of active sodium transport, even with additional ionic currents, a highly active cell will experience significant pump currents. We hypothesize that the presence of compensatory mechanisms to account for the pump-induced firing-rate adaptation is likely to depend on a cell's function; if firing-rate adaptation is a desired feature (i.e. in *Kueh et al., 2016*), compensatory mechanisms may not have evolved. In cells such as electrocytes, however, we argue that compensation of pump currents may be required.

Many biophysical mechanisms and cell properties could serve as compensatory mechanisms, of which some have been suggested and modeled in this article. Other possible compensatory mechanisms are discussed below.

## Regulatory mechanisms

We discussed four mechanisms that may improve firing-rate control under strong electrogenic Na$^+$/K$^+$-ATPase currents: co-expression of Na$^+$/K$^+$-ATPases and sodium leak channels, extracellular potassium buffering, stronger synaptic coupling, and pump voltage dependence. All of these mechanisms can treat the 'symptoms' of electrogenic Na$^+$/K$^+$-ATPase and could be replaced by any other mechanism that achieves the same effect, that is providing an opposing current, diminishing the deviations from baseline pump currents, increasing the entrainment range of a cell, and limiting the pump activity to specific periods of the spike-generation cycle. Some alternative (incomplete list of) compensatory mechanisms that achieve the same effects are discussed below.

Opposing currents do not necessarily have to stem from a sodium leak current, but could also be achieved by other depolarizing currents, such as h-currents (*Roth and Hu, 2020*) or calcium currents (*Bean, 2007*), or through the (relatively small) current that results from co-transport of H$^+$ ions by the Na$^+$/K$^+$-ATPase itself (*Vedovato and Gadsby, 2014*). The former has previously been

shown to counterbalance pump currents in the leech central pattern generator neurons (*Kueh et al., 2016*). Accordingly, the sodium leak also does not have to result from voltage-agnostic channels, but could, for example, be facilitated by a decreased mean half-activation voltage of voltage-dependent sodium channels, which could be achieved by transcriptional regulation of channels with different splice forms (*Liu et al., 2008*). We lastly note that even though additional supposedly 'wasteful' sodium currents might serve a secondary purpose of balancing out fluctuating currents produced via sodium-coupled transport of metabolites (*Berndt and Holzhütter, 2013*).

As we showed, deviations in pump currents, resulting from the susceptibility of the $Na^+/K^+$-ATPase to changes in firing rate, can be diminished by prolonging the timescale of the pump's feedback on cell activity, for example via buffering of extracellular potassium. Similar effects can be obtained from volume increases in intra- and extracellular space or a weaker ion-concentration dependence of the pump. As we showed, for communication with chirps, an uncompensated pump could result in (undesired) spontaneous spiking of electrocytes. In these cases, an increase in the input current required to reach threshold (mediated by additional leak channels) can suppress such activity. Interestingly, compensation could also be provided on the behavioral level: First, an appropriate timing of communication signals, specifically an alternation between frequency rises and chirps, can alleviate undesired changes in baseline pump current due to their opposite impact on pump activity (i.e. increase versus decrease in pump rate). Second, a limitation of individual and cumulative signal duration and amplitude of chirps or frequency rises, respectively, similarly constrains effects on the pump current. This suggests that not only the evolution of ion channels is relevant for shaping communication signals (*Zakon et al., 2006*) but that also the $Na^+/K^+$-ATPases may have played a significant role.

Finally, we argue that effects of the $Na^+/K^+$-ATPase on neuronal dynamics can be partially avoided if the pump is equipped with an appropriately calibrated voltage dependence. A voltage dependence of the pump has been reported experimentally (*Weer et al., 1988*; *Colina et al., 2010*). Specifically, the pumping process depends on many individual steps, each involving a different time scale and voltage dependence (*Wuddel and Apell, 1995*; *Gadsby et al., 2012*). While it remains unclear whether a pump could exhibit a voltage dependence as ideal as in our thought experiment in the last Results section (in particular with respect to the restriction of its activity to periods of potassium channel activation), at least partial benefits from a modulation of pump activity along some of the qualitative principles described in the thought experiment could be expected. Future experimental investigations on the pump's voltage dependence, in particular in highly active cells, will therefore be of interest to support or reject the hypothesis predicted from our modeling approach. Interestingly, properties of the pump's kinetics and voltage dependence have been reported to be highly adaptable in evolution, showing a large heterogeneity in different tissues (*Garcia et al., 2012*; *Stanley et al., 2015*), including different splice variants (*Blanco and Mercer, 1998*) as well as a regulation via RNA editing (*Colina et al., 2010*).

## Implications for disease

Several neurological diseases (*de Carvalho Aguiar et al., 2004*; *Heinzen et al., 2012*; *Rosewich et al., 2012*; *Demos et al., 2014*) have been linked to mutations in the $\alpha$ subunit of the $Na^+/K^+$-ATPase. The origin of many symptoms observed in these diseases, such as epileptic seizures, lies in pathological neuronal network activity including hyperexcitability and altered oscillatory activity (*Yu et al., 2006*). Our modeling work elucidates one mechanism by which altered pump physiology has detrimental effects on cellular computation and can induce pathological network activity. For example, a mutation associated with the rare neurological disease Alternating Hemiplegia of Childhood (AHC) prevents the $Na^+/K^+$-ATPase co-transport of $H^+$ ions, the latter of which normally mitigates some of the pump's negative effects due to its depolarizing contribution (*Sasaki et al., 2014*; *Vedovato and Gadsby, 2014*). From our work, we can conclude that the negative side effects of the pump on network computation should be exacerbated by this mutation. We also hypothesize that not only direct impairment of the $Na^+/K^+$-ATPase may contribute to pathological electrical activity, but also deficits in the postulated compensatory mechanisms, thus opening up additional points of physiological vulnerability to pathology.

## Other factors of relevance

### Reversal potentials

An increase in cellular activity reduces reversal potentials, lowering the 'driving force' in action potential generation, and hence affects firing rates (*Contreras et al., 2021*); vice versa for decreases in activity. The effects of activity-induced concentration changes on neuronal activity therefore stem from a combination of the change in reversal potentials and in pump currents (*Megwa et al., 2023*). Such effects of the reversal potentials were included in the model, and we note that they only contributed mildly (<5%) to the firing-rate adaptation described.

### ATP availability and hypoxia

In addition to ion concentrations, pump rates depend on ATP (*Soltoff and Mandel, 1984*). Limitations in its availability, which are not uncommon in the weakly electric fish due to its foraging in hypoxic environments (*Crampton, 1998*), can alter pump rates more drastically than firing-induced changes in ionic concentrations. In theory, a drastic reduction in pump rate (and thus in pump current) results in elevated cell excitability followed by a depolarization block (*Figure 1C*, *Behbood et al., 2023*). In reality, in such cases, the EOD of the *E. virescens* only reduces in amplitude but does not significantly change in frequency. The EOD only terminates after very long exposure to annoxia (*Reardon et al., 2011*), suggesting the existence of compensatory mechanisms diminishing the effects of altered pump currents on cellular activity.

### Temperature fluctuations

Weakly electric fish are poikilothermic. Inhabiting affluent streams of the Amazon river (*Crampton, 1998*), with temperature variations of four degrees during a day-night cycle (*Rueda-Delgado et al., 2006*), the effects of temperature on spike generation by voltage-dependent channels (*Alonso and Marder, 2020*; *Yu et al., 2012*; *Roemschied et al., 2014*) and $Na^+/K^+$-ATPases in *E. virescens* need to be well balanced to keep cell firing in the physiological range.

### Spatial effects in electrocytes

Electrocytes are approximately 1 mm long and only excitable on the posterior site (*Ban et al., 2015*). The subcellular localization of $Na^+/K^+$-ATPases is hence likely to modulate the impact of the pump's electrogenicity on cell firing (*Joos et al., 2018*). Pumps located on the posterior side are likely to exhibit a more drastic effect on the firing rate of the electrocyte than those on the anterior side. We also note that potential effects of locally constrained ion concentration changes have, in the absence of data about such distributions, been neglected in our model. Constraining changes in ion concentration to specific subcellular locations could influence our results in both directions, either by local amplification or dampening of ion concentration changes (for example, via efficient local buffering).

Taken together, our study demonstrates substantial effects of the $Na^+/K^+$-ATPase's electrogenicity on voltage dynamics in highly active excitable cells. While this property of the most common pump in the nervous system is assumed to serve as a mechanism preventing overexcitability, we show that it can significantly interfere with cellular voltage dynamics via the immediate effects of its highly variable, hyperpolarizing current, posing a particular challenge for highly active cells like electrocytes but perhaps also any other fast-spiking cell in nervous systems. This ultimately calls for strict regulatory mechanisms and may provide an additional evolutionary explanation for the abundance of differential ion channels and the diversity of pump isoforms expressed in excitable cell membranes to not only serve action potential generation, but also the stabilization of firing in biologically realistic environments.

## Methods

All simulations were done in brian2 (*Stimberg et al., 2019*) with a time step of 0.001 ms. We simulated the weakly electric fish electrocyte model from *Joos et al., 2018*, which we incrementally expanded by adding components relevant for modeling the $Na^+/K^+$ pump and corresponding ion concentration dynamics. All parameters were kept the same as in *Joos et al., 2018*, except for the updated

**Table 1.** Updated model parameters.

| Parameter | Value reported in Joos et al., 2018 | Updated value |
|---|---|---|
| $[Na^+]_{in}^0$ | 1.35 mM | 13.5 mM |
| $g_{Na_{max}}$ | 700 $\mu S$ | 900 $\mu S$ |
| $T$ | not reported | 293.15 K |
| $\omega_{in}$ | n. a. | 0.434 mm$^3$ (**Ban et al., 2015**) |
| $\frac{\omega_{in}}{\omega_{out}}$ | n. a. | 2 |
| $v_{onset}$ | n. a. | –76 mV |
| $\epsilon$ | n. a. | 1 |

and additional parameters (corresponding to additional equations presented in the following section) reported in *Table 1*.

The initial (and baseline) intracellular sodium concentration, $[Na^+]_{in}^0$, was updated to attain the physiologically relevant sodium reversal potential of 55 mV, and the maximum sodium conductance, $g_{Na_{max}}$, was updated to facilitate a peak amplitude of 13 mV for the mean frequency of the physiological firing range of 200–600 Hz. Model equations were also kept the same as in *Joos et al., 2018*, except for $I_{stim}$ (Equation 7 in *Joos et al., 2018*), which was adjusted to correct the directions of current flows through acetylcholine receptor (AChR) channels such that

$$I_{stim} = \epsilon syn_{clamp}(t)(-I_{AChRNa} - I_{AChRK}). \tag{1}$$

Note that similarly to *Joos et al., 2018*, the ion flux of calcium ions through AChR channels was neglected under the assumption that similarly to the weakly electric wave-type *Sternopygus macrurus* electrocyte, calcium currents do not significantly contribute to action potentials (*Ferrari and Zakon, 1993*).

Furthermore, the synapse strength, $\epsilon$, was added as a separate parameter. The additional equations used to model ion concentrations and their derivations are described below in more detail. Note that these are generic and could be used to expand any point model of an excitable cell where all currents can be separated into sodium and potassium currents.

## Modeling an excitable cell with Na$^+$/K$^+$ pump

First, the Na$^+$/K$^+$-pump current was added to the model, and a suitable co-expression factor between pump density and sodium leak channels was determined in order to counteract the depolarizing effects of the pump current. Then, dynamical equations for the ion concentrations were added and the energetic demand at the baseline firing regime was estimated to tune pump densities to maintain steady state ion concentrations. Lastly, the feedback loop of pump density on ion concentrations was modeled to maintain ion homeostasis in firing regimes that deviate from baseline. Each of these steps is explained below in more detail.

### Modeling the pump current and sodium leak channel co-expression

In previous work on the computational model of the weakly electric fish electrocyte, it was mentioned that there was an intention to include a Na$^+$/K$^+$ pump (*Joos et al., 2018*). This was, however, abstained from as the author noted that adding an additional current, $I_{pump}$, would leave the model inexcitable. We identified the effects of this pump current on electrocyte excitability and propose additional currents to counteract these effects.

The Na$^+$/K$^+$ pump uses one ATP molecule to exchange three intracellular Na$^+$ ions for two extracellular K$^+$ ions (*Guyton and Hall, 1986*). This leads to a net outflux of one positive ion every time the Na$^+$/K$^+$ pump performs an ion exchange. This net outflux is modeled as an additional current term in the membrane potential evolution equation,

$$C\frac{dv}{dt} = -I_{NaT} - I_{NaP} - I_K - I_L - I_{pump} + I_{stim}(t), \tag{2}$$

which is the 'master equation' that is used for all simulations shown in this study. In the present model, the Na$^+$/K$^+$ pump strength and thus $I_{pump}$ does not depend on the membrane voltage, but on the intra- and extracellular ion concentrations (*Hübel and Dahlem, 2014*). As ion concentration dynamics evolve on much longer timescales than the membrane potential, $I_{pump}$ can in this case be assumed to be approximately constant on the timescale of the membrane potential.

When creating the firing-rate ($r$) vs. input ($\overline{I_{stim}}$) curve ($f$-$I$ curve) of the electrocyte, where $r = f(\overline{I_{stim}})$, an additional outward pump current creates a horizontal translation of $r$, as $r = f(\overline{I_{stim}} - I_{pump})$. In other words, more inward $\overline{I_{stim}}$ is needed to balance out the outward pump current and push the electrocyte to a firing regime.

To counterbalance the hyperpolarizing outward current of the electrogenic Na$^+$/K$^+$ pump, we introduce an additional inward current. As the first approximation of the Na$^+$/K$^+$-ATPase current is constant, we modeled this balancing inward current as a relatively constant leak current

$$I_{NaL} = g_{NaL}(v - E_{Na}), \tag{3}$$

and redefined the leak term in *Equation 2* as

$$I_L = I_{NaL} + I_{KL}, \tag{4}$$

where $I_{KL}$ represents the outward potassium leak;

$$I_{KL} = g_{KL}(v - E_K). \tag{5}$$

As the original model in *Joos et al., 2018* has a leak term with a reversal potential equal to $E_K$ (and thus presumably only represents potassium ion flux), $I_{KL}$ is the same as the leak term in *Joos et al., 2018*, with the same maximal conductance, $g_{KL}$.

For $I_{NaL}$ to balance out $I_{pump}$, $I_{NaL} = -I_{pump}$ should hold for all $t$. In contrast to $I_{pump}$ however, $I_{NaL}$ is highly varying over time as it is dependent on $v$. One condition we can satisfy is for $I_{NaL}$ to cancel out $I_{pump}$ close to the onset of the $f - I$ curve. This should render the firing onset of the electrocyte unchanged. We furthermore assume that $g_{NaL}$ is adjusted at a larger timescale than $I_{pump}$, and that $g_{NaL}$ is expressed to counteract $I_{pump}$ at a baseline level $I_{pump}^0$. Setting $I_{NaL_{onset}} = I_{pump}^0$ and rearranging to get $g_{NaL}$ as a function of $I_{pump}^0$ gives

$$g_{NaL} = \frac{I_{pump}^0}{v_{onset} - E_{Na}}. \tag{6}$$

Here, $v_{onset}$ is the membrane voltage of the electrocyte just before firing onset. Upon injecting a stimulus of 47nA, we find that $v_{onset}$ = -81 mV. After tuning $g_{NaL}$ to counteract the effect of $I_{pump}^0$ using *Equation 6*, we find that the $f$-$I$ curve is least affected by $I_{pump}^0$ when slightly increasing $v_{onset}$ to –76 mV. The properties of sodium leak channel co-expression are exemplified in *Figure 1E–G* and implemented to compensate for baseline pump currents in *Figure 1B, H-J* (with pump), *Figures 2, 4–6* (constant pump), *Appendix 1—figure 1A* (with co-expression), and *Appendix 1—figure 1B–D* (constant pump).

## Modeling dynamic ion concentrations and deriving required pump densities for steady-state ion concentrations

The function of the Na$^+$/K$^+$ pump is to maintain intra- and extracellular ion concentrations at fixed levels. If there were no Na$^+$/K$^+$ pump in an excitable cell, sodium ions would accumulate inside the cells and potassium ions would accumulate in extracellular space. This would reduce the concentration differences between ions in intra- and extracellular space, which impedes the firing of action potentials. The goal of the Na$^+$/K$^+$ pump is therefore to retain fixed sodium and potassium reversal potentials by maintaining ion homeostasis. This is achieved when the energetic supply, which would be the rearranging of ions by the Na$^+$/K$^+$ pump, exactly equals the energetic demand, which is the ion displacement caused by the action potentials.

To fully understand the influence of the Na$^+$/K$^+$ pump on cell excitability, we modeled the ion displacements of action potentials and the pump explicitly (implemented to create *Figures 2, 4–6D*). To this end, we added ion concentration dynamics of intra- and extracellular sodium and potassium to the model equations (*Equations 7–10*) similarly to *Hübel and Dahlem, 2014*, and introduced a

dependency of the reversal potentials on these ion concentrations via the Nernst equation (*Equations 11; 12*):

$$\frac{d[\text{Na}^+]_{\text{in}}}{dt} = \frac{-I_{\text{NaT}} - I_{\text{NaP}} - I_{\text{NaL}} - \epsilon\text{syn}_{\text{clamp}}(t)I_{\text{AChRNa}} - 3I_{\text{pump}}}{F\omega_{\text{in}}}, \tag{7}$$

$$\frac{d[\text{K}^+]_{\text{in}}}{dt} = \frac{-I_{\text{K}} - I_{\text{KL}} - \epsilon\text{syn}_{\text{clamp}}(t)I_{\text{AChRK}} + 2I_{\text{pump}}}{F\omega_{\text{in}}}, \tag{8}$$

$$[\text{Na}^+]_{\text{out}} = [\text{Na}^+]_{\text{out}}^0 - \frac{\omega_{\text{in}}}{\omega_{\text{out}}}([\text{Na}^+]_{\text{in}} - [\text{Na}^+]_{\text{in}}^0), \tag{9}$$

$$[\text{K}^+]_{\text{out}} = [\text{K}^+]_{\text{out}}^0 - \frac{\omega_{\text{in}}}{\omega_{\text{out}}}([\text{K}^+]_{\text{in}} - [\text{K}^+]_{\text{in}}^0), \tag{10}$$

$$E_{\text{Na}} = \frac{RT}{zF} \ln \frac{[\text{Na}^+]_{\text{in}}}{[\text{Na}^+]_{\text{out}}}, \tag{11}$$

$$E_{\text{K}} = \frac{RT}{zF} \ln \frac{[\text{K}^+]_{\text{in}}}{[\text{K}^+]_{\text{out}}}. \tag{12}$$

Here, $F$ is the Faraday constant and $\omega_{\text{in}}$ is the intracellular volume which, as measured by *Ban et al., 2015* is roughly 0.434 mm³. $\frac{\omega_{\text{in}}}{\omega_{\text{out}}}$ is the ratio between the volumes of the intra- and extracellular space. As electrocytes are relatively large compared to their environment, we assumed $\frac{\omega_{\text{in}}}{\omega_{\text{out}}}$ to be 2. Initial and steady-state ionic concentrations in intra- and extracellular space, $[\text{Na}^+]_{\text{in}}^0$, $[\text{K}^+]_{\text{in}}^0$, $[\text{Na}^+]_{\text{out}}^0$, and $[\text{K}^+]_{\text{out}}^0$, were set to 13.5 mM (see *Table 1*), 89 mM, 120 mM, and 2.16 mM (see values reported in *Joos et al., 2018*) respectively. These values were also used to initialize simulations. Note that $I_{\text{stim}}(t)$ has been decomposed into $-\epsilon\text{syn}_{\text{clamp}}(t)I_{\text{AChRNa}}$ and $-\epsilon\text{syn}_{\text{clamp}}(t)I_{\text{AChRK}}$. This has been done to separately track the sodium and potassium displacement caused by the input stimulus.

To simplify the analysis, the model equations were rewritten to cancel out the state variable $[\text{K}^+]_{\text{in}}$, and have the model depend only on $[\text{Na}^+]_{\text{in}}$. To achieve this, we first rewrote the membrane potential equation, *Equation 2*, so that it is separable into Na⁺ and K⁺ currents, by inserting *Equations 1; 4*;

$$C\frac{dv}{dt} = -I_{\text{NaT}} - I_{\text{NaP}} - I_{\text{NaL}} - \epsilon\text{syn}_{\text{clamp}}(t)I_{\text{AChRNa}} - I_{\text{K}} - I_{\text{KL}} - \epsilon\text{syn}_{\text{clamp}}(t)I_{\text{AChRK}} - I_{\text{pump}}. \tag{13}$$

According to *Equations 7; 8*, the membrane potential equation can thus be expressed as

$$C\frac{dv}{dt} = F\omega_{\text{in}} \left( \frac{d[\text{Na}^+]_{\text{in}}}{dt} + \frac{d[\text{K}^+]_{\text{in}}}{dt} \right). \tag{14}$$

Integrating on both sides gives us

$$\frac{C\Delta v}{F\omega_{\text{in}}} = \Delta[\text{Na}^+]_{\text{in}} + \Delta[\text{K}^+]_{\text{in}}. \tag{15}$$

As the membrane conductance $C$ is small, the Faraday constant $F$ is very big, and the intracellular volume $\omega_{\text{in}}$ is also relatively big, we can approximate $\frac{C\Delta v}{F\omega_{\text{in}}} \approx 0$. This proves that in our model, the macroscopic changes in intracellular ion concentrations can always be related by

$$\Delta[\text{Na}^+]_{\text{in}} \approx -\Delta[\text{K}^+]_{\text{in}}. \tag{16}$$

This allows us to rewrite *Equations 8; 10* to depend only on state variable $[\text{Na}^+]_{\text{in}}$

$$[\text{K}^+]_{\text{in}} = [\text{K}^+]_{\text{in}}^0 - ([\text{Na}^+]_{\text{in}} - [\text{Na}^+]_{\text{in}}^0), \tag{17}$$

$$[\text{K}^+]_{\text{out}} = [\text{K}^+]_{\text{out}}^0 + \frac{\omega_{\text{in}}}{\omega_{\text{out}}}([\text{Na}^+]_{\text{in}} - [\text{Na}^+]_{\text{in}}^0). \tag{18}$$

With a perfectly working Na⁺/K⁺ pump, the macroscopic change in intracellular sodium $\Delta[\text{Na}^+]_{\text{in}}$ is zero, which signifies that the energetic supply of the pump exactly equals the energetic demand of the action potentials. From *Equations 9; 17; 18*, we can conclude that if there is no macroscopic change in intracellular sodium, extracellular sodium concentrations and intra- and extracellular potassium concentrations also remain constant.

**Table 2.** All baseline stimuli and tuned parameters presented in this article.

$g_{Na_{max}}$ was tuned to maintain a spike amplitude of 13 mV, and $I^0_{pump}$ and $[Na^+]^{t=0}_{in}$ were tuned to maintain ion homeostasis (and thus steady state values) for the specified stimuli. Empty cells correspond to standard parameter values as reported in *Joos et al., 2018* and *Table 1*.

| Figure | Stimulus (at baseline) | | Tuned parameters | | | |
|---|---|---|---|---|---|---|
| | $syn_{clamp}$ [arb. u.] | $r_{pn}$ [Hz] | $\epsilon$ [arb. u.] | $g_{Na_{max}}$ [$\mu S$] | $I^0_{pump}$ [$\mu A$] | $[Na^+]^{t=0}_{in}$ [mM] |
| *Figures 1H-J and 6A-B* (left) | 0.13 | - | - | - | 1.888 | - |
| *Figure 2* | 0.10 | - | - | - | 1.630 | - |
| *Figure 4* | - | 600 | - | 1300 | 4.074 | - |
| *Figure 5B-D* | - | 260 | 0.5 | 881 | 1.368 | - |
| *Figure 5E-G* | - | 260 | - | 763 | 1.258 | - |
| *Figure 6A-B* (right) | 0.13 | - | - | - | - | - |
| *Figure 6D* | 0.10 | - | - | - | - | 13.517 |

Under the assumption that the Na$^+$/K$^+$ pump is solely responsible for all active sodium transport, long-term ion homeostasis in a high-frequency firing electrocyte can be obtained if we tune the baseline pump current, $I^0_{pump}$, so that $\Delta[Na^+]_{in}$ (*Equation 7*) is zero,

$$I^0_{pump} = \frac{1}{3}\sum \overline{I_{Na}},$$ (19)

where $\sum \overline{I_{Na}}$ is the sum of the time average of all sodium currents, which in this model are $I_{NaT}$, $I_{NaP}$, $I_{NaL}$, and $\epsilon syn_{clamp}(t)I_{AChRNa}$. The time average is here taken in a tonic firing regime where the averaging window equals one spiking period. As $\sum \overline{I_{Na}}$ depends on $I^0_{pump}$ due to the co-expression of pumps and sodium leak channels (*Equation 6*), we iteratively recompute $I^0_{pump}$ according to *Equation 19* until the condition is satisfied with an error margin of 1 nA. This was done to arrive at currents and APs shown in *Figure 1B, H-J* (with pump) and *Figure 6A-C* (constant pump). This procedure was also used to initialize simulations at steady state values in *Figures 2, 4 and 5*. Steady-state pump currents for various stimulus protocols are reported in *Table 2*.

## Modeling the feedback loop of ion concentrations on pump density

Assuming that pump densities are tuned to sustain a fixed baseline firing rate, deviations from this baseline firing will lead to a mismatch between the ion displacement caused by action potentials and the ion restoration of the Na$^+$/K$^+$ pump. This will lead to a shift in intra- and extracellular ion concentrations. As the pump rate and thus $I_{pump}$ is a function of intra- and extracellular ion concentrations (*Hübel and Dahlem, 2014*), the pump rate will adjust accordingly. We model the dependency of $I_{pump}$ on intracellular sodium concentrations $[Na^+]_{in}$ and extracellular potassium concentrations $[K^+]_{out}$ similarly to *Hübel and Dahlem, 2014*,

$$I_{pump} = \frac{4I^0_{pump}}{\left(1 + e^{\frac{[Na^+]^0_{in} - [Na^+]_{in}}{3}}\right)\left(1 + e^{[K^+]^0_{out} - [K^+]_{out}}\right)}.$$ (20)

For simplicity, we adjusted the terms within the exponents so that $I_{pump} = I^0_{pump}$ when $\Delta[Na^+]_{in} = 0$. Here, $I^0_{pump}$ is the pump current that is tuned to facilitate ion homeostasis at the baseline firing rate. As the pump current saturates at $4I^0_{pump}$, which is proportional to the number of Na$^+$/K$^+$-ATPases that are expressed, the baseline pump current $I^0_{pump}$ is also proportional to the pump density. A shift in $I_{pump}$, which can now happen as a consequence of deviations from baseline firing, without co-expression of

$g_{\text{NaL}}$, which is unlikely on small timescales, will lead to a shift in cell excitability. This feedback loop is implemented and its effects shown in *Figures 2, 4–5, and 6D*.

## Mechanisms that improve entrainment

The feedback loop of ion concentrations on pump density alters pump currents when inputs change (*Figure 2*), and consequently disrupts synchronization (*Figures 4 and 5*). Two mechanisms that alleviate these consequences of pump activity on synchronization were explored: extracellular potassium buffering and increased synaptic weights.

Extracellular potassium buffering was implemented to decrease the variation in $\Delta I_{\text{pump}}$, and thereby the variation in mean-driven electrocyte properties (*Equation 20*, *Figure 4D–E*). Transient buffer effects were neglected, and an instantaneous potassium buffer with infinite capacity was assumed by setting

$$[\text{K}^+]_{\text{out}} = [\text{K}^+]_{\text{out}}^0. \tag{21}$$

Synaptic weights were implemented by setting $\epsilon = 0.5$ for weak coupling (*Figure 5B–D*), and $\epsilon = 1$ for strong coupling (*Equation 1*, *Figure 5E–G*).

## Modeling an optimal voltage dependence of the pump

Dynamics of action potential firing would be unaffected by the presence of voltage-dependent electrogenic pumps if the membrane voltage would modulate their activity in a way that the pump current mimics hyperpolarizing voltage-gated and leaky potassium currents. We substantiate this idea by modeling a voltage dependence of the pump that copies the dynamics of potassium currents (*Figure 6*, voltage-dependent pump). To achieve this, we rewrite the baseline pump current $I_{\text{pump}}^0$ as a function of the membrane voltage, and a transformation of the membrane voltage that takes into account the history of the membrane voltage ($n$)

$$I_{\text{pump}}^0 = \frac{1}{3}(g_{\text{K}_\text{L}} + g_{\text{K}_{\max}} n^4)(v - E_\text{K}), \tag{22}$$

which is essentially a scaled version of a combination of the equations that describe the voltage-gated and leaky potassium currents. As the pump current now behaves like $\frac{1}{3}$ of the potassium currents, we can reduce the potassium conductances by $\frac{1}{3}$ and still get qualitatively the same APs, through

$$I_\text{K} = \frac{2}{3} g_{\text{K}_{\max}} n^4 (v - E_\text{K}), \tag{23}$$

and

$$I_{\text{K}_\text{L}} = \frac{2}{3} g_{\text{K}_\text{L}} (v - E_\text{K}). \tag{24}$$

By restructuring the current and pump equations as such, the pump current should always equal $\frac{1}{2}$ of the potassium currents, which exactly satisfies the energetic demand of the cell. There is, however, a third potassium current, $I_{\text{AChRK}}$, that is activated by neurotransmitter release. As we cannot expect the pumps to also be sensitive to neurotransmitter release, the voltage-dependent pump described in *Equation 22* will pump slightly less than necessary to maintain ionic homeostasis. We therefore let the model run until steady state ion concentrations were reached, which happens in close proximity to the baseline concentration of $[\text{Na}^+]_{\text{in}} = 13.500$ mM, which is $[\text{Na}^+]_{\text{in}} = 13.517$ mM. We then initialize subsequent simulations with $[\text{Na}^+]_{\text{in}}^{t=0} = 13.517$ mM.

## Stimulus protocols

In this study, the influence of the pump current on cell excitability was studied for increasingly complex physiologically relevant stimulus protocols. For each stimulus protocol, baseline pump currents, $I_{\text{pump}}^0$, or initial intracellular sodium concentrations, $[\text{Na}^+]_{\text{in}}^0$, were tuned to facilitate ionic homeostasis. Furthermore, when simulating communication paradigms, voltage-gated sodium conductances, $g_{\text{Na}_{\max}}$, were tuned to maintain a spike amplitude of 13 mV. Stimuli and tuned parameters for each

experiment shown in this article are reported in *Table 2* The selected stimuli and their implementations are explained below in more detail.

## Stimulation in the mean-driven regime

For creating frequency-input curves (*Figures 1C; F, 3C, and 6C*), $I_{stim}(t)$ was replaced with a constant input current, and ionic concentrations were fixed to baseline values to avoid the pump-induced changes in firing rates. The $f - I$ curves shown in this article therefore represent instantaneous firing rates. Representative inputs were estimated and used to show the influence of the pump current on tonic firing (*Figure 1D and G*). These inputs were estimated as follows: we first modeled an electrocyte with a periodic synaptic drive as in *Joos et al., 2018*. The frequency of this drive was set to 400 Hz, which is the mean value of reported EODfs (and thus presumable pacemaker firing rates) of 200–600 Hz (*Hopkins, 1974*). Then, $I_{stim}(t)$ was averaged to obtain the average input current (0.63 µA).

A similar approach was taken to show AP shape, current contributions, and energetic demand for a representative constant input (*Figures 1H–J , and 6A-B*). In order to add synaptic currents, which account for additional in- and outfluxes of sodium and potassium, in these experiments, the electrocyte was stimulated with a constant synaptic drive ($syn_{clamp}$=0.13) taken from the average synaptic drive resulting from a 400 Hz pacemaker stimulus.

To showcase the pump-current induced firing-rate adaptation in *Figure 2*, the baseline synaptic drive was chosen as the mean $syn_{clamp}(t)$ for a periodic drive of 300 Hz ($syn_{clamp}$=0.10). This was chosen so that doubling the synaptic drive would result in a physiologically plausible synaptic drive of 0.18, which would be the average of a 600 Hz periodic drive.

## Periodic stimulation

Next, we showed the impact of pump-current induced firing-rate adaptation on the synchronization of an excitable cell to periodic input. In particular, we studied the entrainment of the electrocyte to the pacemaker nucleus. The pacemaker was not modeled explicitly, but only the time-dependent currents that would result from the excitatory synapse were (as in *Joos et al., 2018*). Here, the parameter that reflects the (unitless) magnitude of the synaptic conductance, $syn_{clamp}(t_{pn})$, is modeled by a piecewise function that resets $t_{pn} \to 0$ at the pacemaker firing rate, $r_{pn}$;

$$t_{pn}(t) = t \ \ \text{mod} \ \frac{1}{r_{pn}}, \tag{25}$$

$$syn_{clamp}(t_{pn}) = \begin{cases} \dfrac{t_{pn}}{0.05 \text{ ms}}, & t_{pn} \leq 0.05 \text{ ms} \\ 1, & 0.05 \text{ ms} < t_{pn} \leq 0.25 \text{ ms} \\ e^{\dfrac{-(t_{pn} - 0.25 \text{ ms})}{0.1 \text{ ms}}}, & t_{pn} > 0.25 \text{ ms}. \end{cases} \tag{26}$$

To demonstrate physiologically relevant scenarios in which a variable $r_{pn}$ affects the pump current, $r_{pn}$ was set to model chirps (cessations in firing) and frequency rises. To ensure a significant effect of firing-rate changes on the pump current, relatively long chirps were initiated in a fish with high baseline firing rates, and relatively high frequency rises in a fish with low baseline firing rates. Baseline firing rates, chirp duration, frequency rise amplitude, and frequency rise timescale are representative of EOD signals found in experimental settings (*Hopkins, 1974*).

To model chirps (*Figure 4*), $r_{pn}$ was set to 600 Hz, and after 100 ms, chirps were generated where $t_{pn}$ was only reset after a period of 20 spikes i.e. $\frac{20}{r_{pn}}$. After this period, $t_{pn}$ was again reset with a frequency $r_{pn}$ for 100 ms. This was repeated 10 times to simulate 10 consecutive chirps.

To model frequency rises (*Figure 5*), $r_{pn}$ was set to 260 Hz, and after 6 s, frequency rises were generated where $r_{pn}$ was set by the following formula

$$r_{pn}(t_{rise}) = r_{rise}e^{-\dfrac{t_{rise}}{\tau_{rise}}} + r_{pn}^0, \tag{27}$$

where $r_{rise}$ is the amplitude of the frequency rise, which was set to 40 Hz, $t_{rise}$ is the elapsed time since the onset of the frequency rise, $\tau_{rise}$ is the timescale of the frequency rise which was set to 1 s, and $r_{pn}^0$

is the baseline frequency which was set to 260 Hz. Frequency rises were initiated every 2 s, 10 times in a row.

## Spike entrainment measure

To quantify the accuracy of entrainment between spikes emitted by the PN at times, $t_1 < ... < t_m$ and spikes emitted by the electrocyte at times $t_1 < ... < t_n$, where $t_n^e < t_m^p$.

Let $j(t) = j : t_j^p < t < t_{j+1}^p$ denote the index of the PN spike interval in which a given electrocyte spike time resides. Then define the phases of electrocyte spikes relative to pacemaker inter-spike intervals as

$$\theta_k = \frac{t_k^e - t_{j(t_k)}^p}{t_{j(t_k)+1}^p - t_{j(t_k)}^p}. \tag{28}$$

The rigidity of the entrainment phase is then quantified by the circular variance as the mean resultant length

$$\bar{R} = \frac{1}{n} \left| \sum_{k=1}^{n} \exp i 2\pi \theta_k \right|. \tag{29}$$

This score, which is referred to as the entrainment index, is shown on top of spike trains in *Figure 5B–G*.

## Acknowledgements

We thank Dr. Louisiane Lemaire and Mahraz Behbood for fruitful discussions. This project has received funding from the Einstein Foundation Berlin (grant number EP-2021–621).

## Additional information

### Funding

| Funder | Grant reference number | Author |
|---|---|---|
| Einstein Stiftung Berlin | EP-2021-621 | Susanne Schreiber |

The funders had no role in study design, data collection and interpretation, or the decision to submit the work for publication.

### Author contributions

Liz Weerdmeester, Conceptualization, Software, Methodology, Writing – original draft, Writing – review and editing; Jan-Hendrik Schleimer, Conceptualization, Supervision, Methodology, Writing – original draft, Writing – review and editing; Susanne Schreiber, Conceptualization, Supervision, Funding acquisition, Writing – original draft, Writing – review and editing

### Author ORCIDs

Liz Weerdmeester ⓘ https://orcid.org/0009-0005-3024-4086
Jan-Hendrik Schleimer ⓘ https://orcid.org/0000-0002-2156-330X
Susanne Schreiber ⓘ https://orcid.org/0000-0003-3913-5650

Reviewer #1 (Public review): https://doi.org/10.7554/eLife.103781.4.sa1
Reviewer #2 (Public review): https://doi.org/10.7554/eLife.103781.4.sa2
Author response https://doi.org/10.7554/eLife.103781.4.sa3

# Additional files

## Supplementary files
MDAR checklist

Source code 1. All results presented in this article can be reproduced via this source code.

## Data availability
All results presented in this article can be reproduced via the code appended in Source Code 1, and is also published at https://itbgit.biologie.hu-berlin.de/compneurophys_pub/electrocyte_nakatpase.

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

# Appendix 1

## Main results are independent from synaptic drive

As previously shown in *Joos et al., 2018*, AP amplitude decreases with increasing input currents (*Appendix 1—figure 1A*, left). This effect remains upon addition of either a pump with constant pump rate and co-expressed sodium leak channels (*Appendix 1—figure 1A*, center), or a voltage-dependent pump (*Appendix 1—figure 1A*, right). Interestingly, even though the shape of the current contributions (*Appendix 1—figure 1B*) and the APs (*Appendix 1—figure 1C*) look very different for low (*Appendix 1—figure 1C*, top) and high (*Appendix 1—figure 1C*, bottom) inputs, the total sodium and potassium displacement per AP, and thus the pump rate, is roughly the same (*Appendix 1—figure 1D*). Under the assumption that voltage-gated sodium channel (NaV) expression is adjusted to facilitate fixed AP amplitudes, however (as in *Joos et al., 2018*), more NaV channels would be expressed in fish with higher synaptic drives. This would then result in additional sodium influx per AP and result in higher energetic requirements per AP for electrocytes with higher firing rates (as also shown in *Joos et al., 2018*).

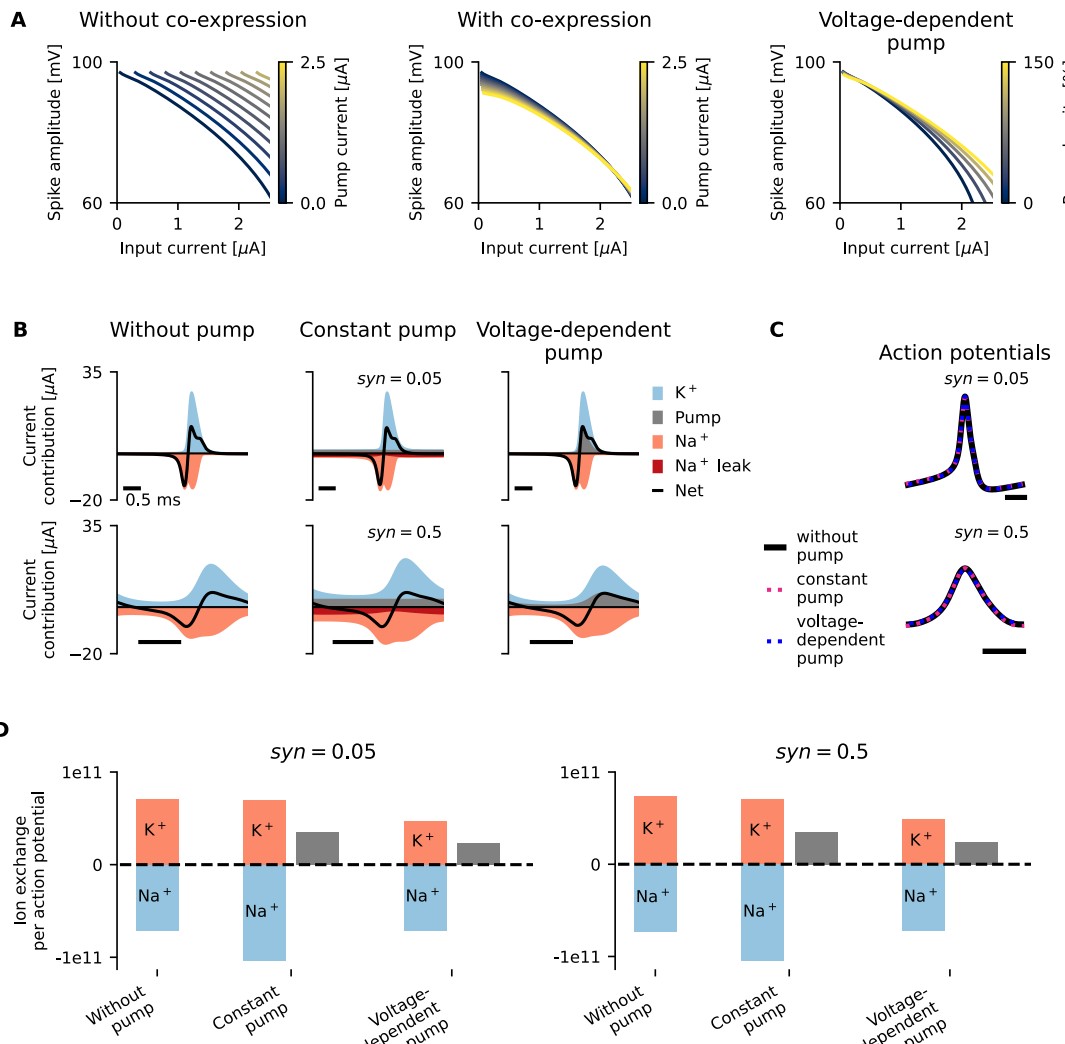

**Appendix 1—figure 1.** Spike amplitudes decrease with increasing input current, but current contributions per action potential (AP) remain the same. (**A**) Spike amplitudes decrease with increasing input currents. Without any compensatory mechanisms (i.e. co-expression of sodium leak channels, left), the pump current shifts the influence of input current on spike amplitude similarly to that on the frequency-vs-input curve (*Figure 1C*). Pump currents that are regulated either through co-expressed channels (center) or a voltage dependence of the pump (right) have little influence on the relation between input current and spike amplitude. (**B**) A comparison of all currents
*Appendix 1—figure 1 continued on next page*

*Appendix 1—figure 1 continued*

that contribute to the AP between constant small (top, syn = 0.05), and large (bottom, syn = 0.5) synaptic inputs which result in low and high firing rates respectively. (**C**) AP shapes of electrocyte models without and with a constant or voltage-dependent pump current are indistinguishable. (**D**) Ion exchanges per AP, and thus pump rates, are similar for low (left) and high (right) synaptic inputs.

# Appendix 2

## Phase oscillator theory to quantify entrainment

The observation that each PN spike typically causes one electrocyte spike suggests that electrocytes are excitable cells, kicked above threshold by synaptic PN inputs and then return to rest. However, taking into account the sustained high frequency PN firing rates, $r_{pn}$, and the kinetics of the electrocyte acetylcholine receptor shows that the model is better understood as a mean-driven entrained oscillator. To see this, the membrane equation of the electrocyte, *Equation 2*, is rewritten to separate the stimulus current into a time-averaged DC component, *Equation 30*, and a time-dependent zero-mean component.

Averaging the stimulus over one PN period gives

$$\overline{I_{\text{stim}}} = r_{pn} \int_0^{\frac{1}{r_{pn}}} I_{\text{stim}}(t)\, dt. \tag{30}$$

As will be shown below, in many cases, the mean drive, $\overline{I_{\text{stim}}}$, is sufficient to induce spiking the electrocyte. Reformulating *Equation 2* to include the mean stimulus expressed in *Equation 30* reads

$$C\frac{dv}{dt} = \underbrace{-I_{\text{NaT}} - I_{\text{NaP}} - I_K - I_L - I_{\text{pump}} + \overline{I_{\text{stim}}}}_{\text{mean-driven electrocyte}} + \underbrace{(I_{\text{stim}}(t) - \overline{I_{\text{stim}}})}_{\text{time dependent PN input}}. \tag{31}$$

As the baseline pump current, which depends on the mean input (see *Equation 19*, Methods), is compensated for through the co-expression of sodium leak channels to generate similar mean-driven dynamics for different baseline pump currents (see *Figure 1F* and *Equation 6*, Methods), the baseline pump current, $I_{\text{pump}}$, is here set to zero to simplify analysis. Note that mean-driven properties will be qualitatively similar, but slightly different depending on the baseline pump current, and thus the co-expressed sodium leak channels (as also shown in slightly varying fI curves in *Figure 1F*).

We can now characterize the mean-driven electrocyte by the relation of its firing rate to mean input in terms of its $f$-$I$ curve (*Figure 3C*). We can furthermore determine the time-averaged stimulus for various PN driving frequencies (*Figure 3B*). When comparing the mean-driven frequency of the electrocyte, $r_e$, to the PN driving frequency, $r_{pn}$, we find mismatches in frequencies that can go up to 180 Hz (*Figure 3D*). As proven by *Joos et al., 2018*, the electrocyte model can be entrained by $r_{pn}$ ranging from 200 to 600 Hz. We can therefore assume that the input stimulus is strong enough to overcome large frequency mismatches between $r_e$ and $r_{pn}$.

To quantify the allowed frequency mismatches between $r_e$ and $r_{pn}$ for synchronization, we treat the mean-driven electrocyte as a periodic oscillator that is to be entrained by an external force, which is the PN. The evolution of the phase of the electrocyte can now be expressed as

$$\frac{d\phi}{dt} = r_e + Z(\phi)x(t). \tag{32}$$

Here, $r_e$ is the frequency of the mean-driven electrocyte, $Z(\phi)$ is the change in phase of the electrocyte evoked by perturbations of the membrane potential, and $x(t)$ is the time-dependent perturbation caused by the zero mean pacemaker input stimulus

$$x(t) = \frac{1}{C}(I_{\text{stim}}(t) - \overline{I_{\text{stim}}}). \tag{33}$$

One can now define a variable that describes the phase difference between the electrocyte and the PN as

$$\psi = \phi - r_{pn}t. \tag{34}$$

If there exists a phase $\psi$ for which $\psi$ does not change over time i.e. $\frac{d\psi}{dt} = 0$, the electrocyte and the PN will be phase-locked, or entrained. The equation for evolution of the phase difference is obtained by combining and rearranging *Equation 34* and *Equation 32*

$$\frac{d\psi}{dt} = r_e - r_{pn} + Z(\psi + r_{pn}t)x(t). \tag{35}$$

As $\psi \ll r_{pn}t$ is a slow variable, one period of $\psi$ will `see' a lot of PN periods. Hence, the method of averaging yields

$$\frac{d\psi}{dt} = r_e - r_{pn} + r_{pn} \int_0^{\frac{1}{r_{pn}}} Z(\psi + r_{pn}t)x(t)\, dt. \tag{36}$$

From this equation, the minimum and maximum $r_{pn}$ for which the electrocyte and the PN can be phase-locked is defined as

$$r_{pn_{max}} = r_e + r_{pn} \max_{0 \leq \psi \leq 1} \int_0^{\frac{1}{r_{pn}}} Z(\psi + r_{pn}t)x(t)\, dt, \tag{37}$$

and

$$r_{pn_{min}} = r_e + r_{pn} \min_{0 \leq \psi \leq 1} \int_0^{\frac{1}{r_{pn}}} Z(\psi + r_{pn}t)x(t)\, dt. \tag{38}$$

The range $[r_{pn_{min}}, r_{pn_{max}}]$ is referred to as the entrainment range.

The mean-driven features $r_e$ and $Z(\phi)$ are altered upon a deviation of the pump current $\Delta I_{pump}$ (*Appendix 2—figure 1A, B*), which is in essence the same as a deviation in $\overline{I_{stim}}$. $\Delta I_{pump}$ thus affects $[r_{pn_{min}}, r_{pn_{max}}]$ and thereby the entrainment of the pacemaker-driven electrocyte (*Appendix 2—figure 1C*).

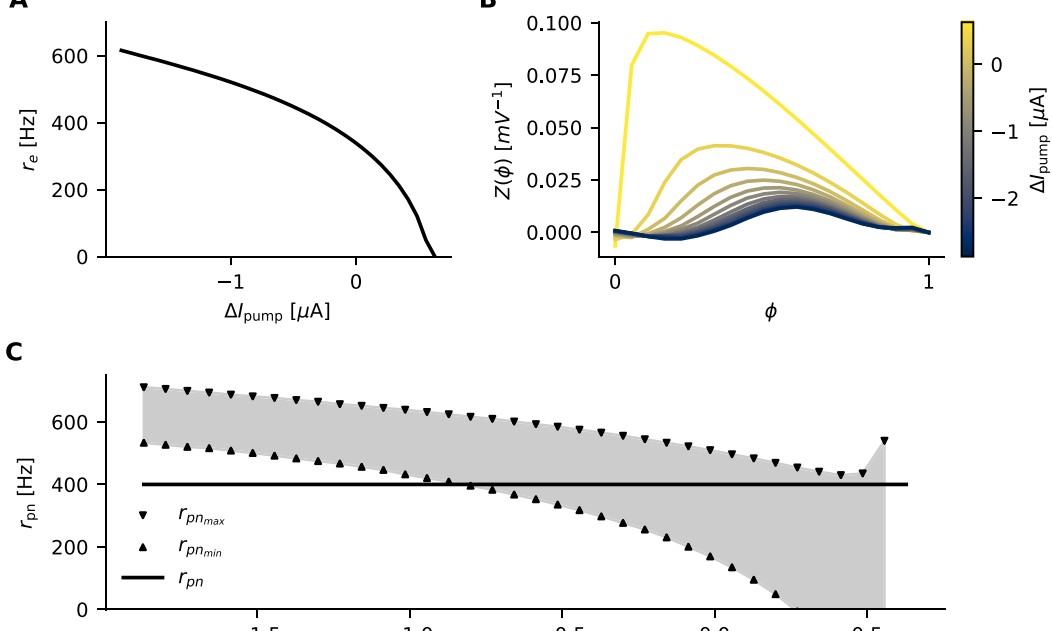

**Appendix 2—figure 1.** A deviation in pump current $\Delta I_{pump}$ alters mean-driven electrocyte properties and thereby its entrainment region. (**A**) Mean-driven electrocyte frequency $r_e$ as a function of $\Delta I_{pump}$. (**B**) Phase Response Curves (PRCs, $Z(\phi)$) as a function of $\Delta I_{pump}$. (**C**) The entrainment range, $[r_{pn_{min}}, r_{pn_{max}}]$ which is a function of mean-driven electrocyte properties (**A**, **B**, *Equations 37; 38*), changes upon deviations in pump current. For very strong deviations in $I_{pump}$, the pacemaker frequency $r_{pn}$ falls out of the entrainment range, which means that the electrocyte will not lock to the pacemaker in this regime.

The influence of the synaptic weight on entrainment (*Figure 5B–G*) can also be explained by equations (*Garcia et al., 2012*; *Gadsby et al., 2012*), as it increases x(t) and therefore $[r_{pn_{min}}, r_{pn_{max}}]$.

## Phase Response Curves

To solve *Equations 37; 38*, PRCs need to be computed. PRCs were obtained for various $\overline{I_{stim}}$ by first injecting constant input current $\overline{I_{stim}}$ to the electrocyte. Then, the phase of the electrocyte was defined as the peak of one spike in the tonically firing electrocyte to the next peak. The membrane voltage was then perturbed by 1 mV at 20 different phases linearly interpolated between 0 and 1, and the resulting time delays or advances of the next spike was recorded as the phase response.

# Appendix 3

## Relation between integrated sodium and potassium currents over one ISI

In a conductance-based model without pump, where all currents are sodium or potassium based (which is the case in the model used for this study, as shown in *Equation 13*, if the pump current is set to zero), the cumulative sodium and potassium currents have to add up to zero between two spikes. This becomes evident when expressing the membrane equation as such;

$$C\frac{dv}{dt} = -I_{\mathrm{Na}} - I_{\mathrm{K}}. \tag{39}$$

If $v$ is in a limit cycle with length $T$,

$$v(t) = v(t + T)\forall t,$$

and

$$v(t) - v(t + T) = 0\forall t. \tag{40}$$

If we integrate the voltage equation (*Equation 39*) on both sides from $t$ to $t + T$, we get

$$C[v(t + T) - v(t)] = -\int_t^{t+T} I_{\mathrm{Na}}(t)dt - \int_t^{t+T} I_{\mathrm{K}}(t)dt. \tag{41}$$

If we plug in *Equation 40*, we get

$$\int_t^{t+T} I_{\mathrm{Na}}(t)dt + \int_t^{t+T} I_{\mathrm{K}}(t)dt = 0, \tag{42}$$

which means that the integrated sodium and potassium currents over the time course of one ISI should always be equal to each other.

# Appendix 4

## Considerations on metabolic costs of compensatory mechanisms

Quantitative estimates of metabolic costs in this study are based on the ATP that is required to fuel the $Na^+/K^+$ pump. This includes the cost of the restoration of sodium and potassium ions that flow to support action potentials, resting potentials, and postsynaptic potentials.

The co-expression of pumps and sodium leak channels (see *Figure 1*) and even an ideal voltage dependence of the pump (see *Figure 6*) have a direct impact on the metabolic cost related to this ATP-fueled $Na^+/K^+$ pump. By integrating the net pump current over time and dividing by one elemental charge, we find the rate of ATP that is consumed for either compensatory mechanism. When compensating a relatively `constant' $Na^+/K^+$-pump current with sodium leak channels, the amount of ATP spent on pumping sodium is 33% higher than it would be for a voltage-dependent pump (see *Equation 22*, Methods).

The impact that either of these compensatory mechanisms has on the whole cell, however, also depends on other costs, such as those related to cellular maintenance. A voltage-dependent pump would save costs related to $Na^+/K^+$ pumping, which, based on energy budgets formerly estimated for AP-firing neurons in the brain (*Howarth et al., 2012*), is likely to be one of the main contributors to the total metabolic cost (in cerebellar cortex, for example, amounting to >50% of the total metabolic cost). Because the peak load of a voltage-dependent pump, however, is four times higher than a relatively constant pump, four times more $Na^+/K^+$ pumps would need to be expressed on the cell membrane. To be more exact, if a single pump translocates around 450 sodium ions per second (*Gennis, 2013*), $8 \times 10^{10}$ pumps are required to support constant pumping, and $32 \times 10^{10}$ pumps are needed to support voltage-dependent pumping. If one assumes the electrocyte is a perfect cylinder, and its membrane surface were smooth (an approximation not too realistic), the total available membrane space would be 3.4 $mm^2$ (*Ban et al., 2015*). If the $Na^+/K^+$ATPase expression density would be as high as in the outer medulla of rabbit kidney (*Deguchi et al., 1977*), where ATPases are densely packed, a smooth electrocyte membrane would `fit' $4.2 \times 10^{10}$ pumps, which is two times less than necessary for constant pumping, and eight times less than required for voltage-dependent pumps. According to our model, therefore, the invaginations on the posterior side of the membrane (*Ban et al., 2015*) are necessary to drastically increase membrane area in order to support the large number of pumps required for ion restoration. This, in turn, would increase the `housekeeping' costs of the cell related to turnover of macromolecules, axoplasmic transport, and mitochondrial proton leak, which in different brain areas are estimated to occupy 25–50% of the total energy budget (*Kety, 1957*; *Attwell and Laughlin, 2001*). As there is insufficient data on the ratio between costs related to $Na^+/K^+$ pumping and `housekeeping costs', and the fraction of housekeeping costs related to $Na^+/K^+$-pump maintenance, a quantitative comparison of the metabolic cost of the two compensatory mechanisms remains challenging. Future experiments that would aid in answering this question could involve blockage of electrocyte $Na^+/K^+$ pumps and comparing oxygen consumption to a control where electrocyte $Na^+/K^+$ pumps are functional.

Another compensatory mechanism that was discussed in this article is extracellular potassium buffering (see *Figure 4*), which in electrocytes likely occurs via its extensive capillary beds (*Ban et al., 2015*) that transport excess extracellular potassium to the kidney. Assuming that an equal amount of ATP is needed in total to fuel $Na^+/K^+$ pumps, either all in the electrocyte, or partly at the electrocyte and partly in the kidney, the additional costs incurred by the extracellular potassium buffer would be dominated by the structural and maintenance costs of the capillaries. We are, however, not aware of an accurate estimate of these costs, especially since the capillaries also have additional functions such as providing other resources and transporting other waste products.

Lastly, a strong synapse was said in the article to support cell entrainment under fluctuating pump currents (see *Figure 5*), but also to incur additional metabolic costs. In the example shown in the main text, however, baseline $Na^+/K^+$ costs are smaller for a stronger synapse; see *Figure 5B* (weak synapse) vs. *Figure 5E* (strong synapse). This is the case because, similarly as shown in Figure 7B in *Joos et al., 2018*, a weak synapse elicits smaller postsynaptic potentials, which lowers the AP peak with respect to a stronger synapse. To make a fair comparison on the metabolic costs between a weak and a strong synapse, voltage-gated sodium conductances were scaled to maintain a peak amplitude of 13 mV (see *Table 2*, Methods). For weak synaptic stimulation, a higher voltage-gated

sodium conductance was needed to reach this peak amplitude, which, due to the excess inflow of sodium through these voltage-gated channels, resulted in an increase of 10% in ATP consumption by Na$^+$/K$^+$ pumps with respect to strong synaptic stimulation.

There are, however, additional costs that scale with synapse strength, such as the restoration of presynaptic calcium, the restoration of (presumably small amounts of) postsynaptic calcium, and neurotransmitter packaging and recycling. In the brain, these costs are estimated to be 0.18–1 times the cost of fueling the Na$^+$/K$^+$ pumps that restore the sodium ions that traverse neurotransmitter receptor channels (*Howarth et al., 2012*; *Liotta et al., 2012*). In our model, merely 11% of sodium ions enter the electrocyte via neurotransmitter receptor channels in the strong-synapse case. Assuming that the above-mentioned additional costs are equal to those related to Na$^+$/K$^+$ pumping of neurotransmitter-related currents (according to the upper bound estimate by *Liotta et al., 2012*), a weak synapse (half the size of the strong synapse) would incur a cost increase of 5.5% and a strong synapse would incur an increase of 11%. This would, however, still result in a 4% higher cost efficiency of a strong synapse compared to a weak synapse.

There is reason to believe that the fraction of the energy budget related to the restoration of presynaptic calcium, the restoration of (presumably small amounts of) postsynaptic calcium, and neurotransmitter packaging and recycling in the electrocyte could differ significantly from those estimated by *Howarth et al., 2012*; *Liotta et al., 2012*. First, to the best of our knowledge, such energy budget estimations have only been done for neurons active at significantly lower firing rates than electrocytes (by a factor of approximately 100), and, second, operate mostly under the glutamate neurotransmitter, while electrocyte receptor channels are activated by acetylcholine. An accurate estimate of the impact of synapse strength on the electrocyte energy budget, therefore, requires quantitative data on the rapid dynamics of acetylcholine production in the presynaptic neuron and recycling in the synaptic cleft, which, currently, is also hard to estimate.

Supported by the above-mentioned considerations, we argue that the impact of mechanisms that compensate for Na$^+$/K$^+$-pump currents on an electrocyte's metabolic cost could be significant. Due to the absence of more detailed experimental quantification, a plausible quantitative cost estimate remains beyond the scope of this article. We note, however, that although the metabolic costs of potassium buffering and synaptic strength are likely to differ between cell types, the energetic estimate of the respective ATP requirements by Na$^+$/K$^+$ pumps for constant vs. voltage-dependent pumping generalizes and extends to all excitable cell types (as is discussed in the Discussion in the main text, see 'Generalization to other cell types').

