## [Editor Report · eLife Assessment]

This **important** study provides new insights into the lesser-known effects of the sodium-potassium pump on how nerve cells process signals, particularly in highly active cells like those of weakly electric fish. The computational methods used to establish the claims in this work are **compelling** and can be used as a starting point for further studies.

---

## [Referee Report · Reviewer #1 (Public review)]

Summary:

The authors aim to explore the effects of the electrogenic sodium-potassium pump (Na+/K+-ATPase) on the computational properties of highly active spiking neurons, using the weakly-electric fish electrocyte as a model system. Their work highlights how the pump's electrogenicity, while essential for maintaining ionic gradients, introduces challenges in neuronal firing stability and signal processing, especially in cells that fire at high rates. The study identifies compensatory mechanisms that cells might use to counteract these effects, and speculates on the role of voltage dependence in the pump's behavior, suggesting that Na+/K+-ATPase could be a factor in neuronal dysfunctions and diseases

Strengths:

(1) The study explores a less-examined aspect of neural dynamics-the effects of Na+/K+-ATPase electrogenicity. It offers a new perspective by highlighting the pump's role not only in ion homeostasis but also in its potential influence on neural computation.

(2) The mathematical modeling used is a significant strength, providing a clear and controlled framework to explore the effects of the Na+/K+-ATPase on spiking cells. This approach allows for the systematic testing of different conditions and behaviors that might be difficult to observe directly in biological experiments.

(3) The study several interesting compensatory mechanisms, such as sodium leak channels and extracellular potassium buffering, which provide useful theoretical frameworks for understanding how neurons maintain firing rate control despite the pump's effects.

Comments on revisions:proposes

The revised manuscript is notably improved.

---

## [Referee Report · Reviewer #2 (Public review)]

Summary:

The paper by Weerdmeester, Schleimer, and Schreiber uses computational models to present the biological constraints under which electrocytes - specialized, highly active cells that facilitate electro-sensing in weakly electric fish-may operate. The authors suggest potential solutions that these cells could employ to circumvent these constraints.

Electrocytes are highly active or spiking (greater than 300Hz) for sustained periods (for minutes to hours), and such activity is possible due to an influx of sodium and efflux of potassium ions into these cells after each spike. The resulting ion imbalance must be restored, which in electrocytes, as with many other biological cells, is facilitated by the Na-K pumps at the expense of biological energy, i.e., ATP molecules. For each ATP molecule the pump uses, three positively charged sodium ions from the intracellular space are exchanged for two positively charged potassium ions from the extracellular space. This creates a net efflux of positive ions into the extracellular space, resulting in hyperpolarized potentials for the cell over time. For most cells, this does not pose an issue, as their firing rate is much slower, and other compensatory mechanisms and pumps can effectively restore the ion imbalances. However, in the electrocytes of weakly electric fish, which spike at exceptionally high rates, the net efflux of positive ions presents a challenge. Additionally, these cells are involved in critical communication and survival behaviors, underscoring their essential role in reliable functioning.

In a computational model, the authors test four increasingly complex solutions to the problem of counteracting the hyperpolarized states that occur due to continuous NaK pump action to sustain baseline activity. First, they propose a solution for a well-matched Na leak channel that operates in conjunction with the NaK pump, counteracting the hyperpolarizing states naturally. Their model shows that when such an orchestrated Na leak current is not included, quick changes in the firing rates could have unexpected side effects. Secondly, they study the implications of this cell in the context of chirps-a means of communication between individual fish. Here, an upstream pacemaking neuron entrains the electrocyte to spike, which ceases to produce a so-called chirp - a brief pause in the sustained activity of the electrocytes. In their model, the authors demonstrate that including the extracellular potassium buffer is necessary to obtain a reliable chirp signal. Thirdly, they tested another means of communication in which there was a sudden increase in the firing rate of the electrocyte, followed by a decay to the baseline. For this to occur reliably, the authors emphasize that a strong synaptic connection between the pacemaker neuron and the electrocyte is necessary. Finally, since these cells are energy-intensive, they hypothesize that electrocytes may have energy-efficient action potentials, for which their NaK pumps may be sensitive to the membrane voltages and perform course correction rapidly.

Strengths:

The authors extend an existing electrocyte model (Joos et al., 2018) based on the classical Hodgkin and Huxley conductance-based models of sodium and potassium currents to include the dynamics of the sodium-potassium (NaK) pump. The authors estimate the pump's properties based on reasonable assumptions related to the leak potential. Their proposed solutions are valid and may be employed by weakly electric fish. The authors explore theoretical solutions to electrosensing behavior that compound and suggest that all these solutions must be simultaneously active for the survival and behavior of the fish. This work provides a good starting point for conducting in vivo experiments to determine which of these proposed solutions the fish employ and their relative importance. The authors include testable hypotheses for their computational models.

---

## [Author Response]

The following is the authors’ response to the previous reviews

**Reviewer #1 (Public review):**
Summary:The authors aim to explore the effects of the electrogenic sodium-potassium pump (Na^+^/K^+^ATPase) on the computational properties of highly active spiking neurons, using the weakly-electric fish electrocyte as a model system. Their work highlights how the pump's electrogenicity, while essential for maintaining ionic gradients, introduces challenges in neuronal firing stability and signal processing, especially in cells that fire at high rates. The study identifies compensatory mechanisms that cells might use to counteract these effects, and speculates on the role of voltage dependence in the pump's behavior, suggesting that Na^+^/K^+^-ATPase could be a factor in neuronal dysfunctions and diseasesStrengths:(1) The study explores a less-examined aspect of neural dynamics-the effects of Na^+^/K^+^-ATPase electrogenicity. It offers a new perspective by highlighting the pump's role not only in ion homeostasis but also in its potential influence on neural computation.(2) The mathematical modeling used is a significant strength, providing a clear and controlled framework to explore the effects of the Na^+^/K^+^-ATPase on spiking cells. This approach allows for the systematic testing of different conditions and behaviors that might be difficult to observe directly in biological experiments.(3) The study proposes several interesting compensatory mechanisms, such as sodium leak channelsand extracellular potassium buffering, which provide useful theoretical frameworks for understanding how neurons maintain firing rate control despite the pump's effects.Weaknesses:(1) While the modeling approach provides valuable insights, the lack of experimental data to validate the model's predictions weakens the overall conclusions.(2)The proposed compensatory mechanisms are discussed primarily in theoretical terms without providing quantitative estimates of their impact on the neuron's metabolic cost or other physiological parameters.Comments on revisions:The revised manuscript is notably improved.

We thank the reviewer for their concise and accurate summary and appreciate the constructive feedback on the article’s strengths and weaknesses. Experimental work is beyond the scope of our modeling-based study. However, we would like our work to serve as a framework for future experimental studies into the role of the electrogenic pump current (and its possible compensatory currents) in disease, and its role in evolution of highly specialized excitable cells (such as electrocytes).

Quantitative estimates of metabolic costs in this study are limited to the ATP that is required to fuel the Na^+^/K^+^ pump. By integrating the net pump current over time and dividing by one elemental charge, one can find the rate of ATP that is consumed by the Na^+^/K^+^ pump for either compensatory mechanism. The difference in net pump current is thus proportional to ATP consumption, which allows for a direct comparison of the cost efficiency of the Na^+^/K^+^ pump for each proposed compensatory mechanism. The Na^+^/K^+^ pump is however not the only ATP-consuming element in the electrocyte, and some of the compensatory mechanisms induce other costs related to cell ‘housekeeping’ or presynaptic processes. We now added a section in the appendix titled ‘Considerations on metabolic costs of compensatory mechanisms’ (section 11.4), where we provide rough estimates on the influence of the compensatory mechanisms on the total metabolic costs of the cell and membrane space occupation. Although we argue that according these rough estimates, the impact of discussed compensatory mechanisms could be significant, due to the absence of more detailed experimental quantification, a plausible quantitative cost estimate on the whole cell level remains beyond the scope of this article.

**Reviewer #1 (Recommendations for the authors):**
I just have a few recommendations on the updated manuscript.(1) When exploring the different roles of Na^+^/K^+^-ATPase in the Results section, the authors employed many different models. For instance, the voltage equation on page 15, voltage equation (2) on page 22, voltage equation (12) on page 24, voltage equation (30) on page 32, and voltage equation (38) on page 35 are presented as the master equations for their respective biophysical models. Meanwhile, the phase models are presented on page 29 and page 33. I would recommend that the authors clearly specify which equations correspond to each subsection of the Results section and explicitly state which equations were used to generate the data in each figure. This would help readers more easily follow the connections between the models, the results, and the figures.

We thank the reviewer for pointing out that the links of the different voltage equations to the results could be expressed more explicitly in the article. All simulations were done using the ‘master equation’ expressed in Eq. 2, and the other voltage equations that are specified in the article (in the new version of the article Eqs. 13, 31, and 39) are reformulations of Eq. 2 to analytically show different properties of the voltage equation (Eq. 2). This has now been mentioned in the article when formulating the voltage equations, and the equation for the total leak current (in the new version Eq. 3) has been added for completeness.

(2) The authors may want to revisit their description and references concerning Eigenmannia virescens. For example, wave-type weakly electric fish (e.g., Eigenmannia) and pulse-type weakly electric fish (e.g., Gymnotus carapo) exhibit large differences, making references 52-55 may be inappropriate for subsection 4.3.1, as these studies focus on Gymnotus carapo. Additionally, even within wave-type species, chirp patterns vary. For example, Eigenmannia can exhibit short "pauses"-type chirps, whereas Apteronotus leptorhynchus (another waver-form fish) does not (https://pubmed.ncbi.nlm.nih.gov/14692494/).

We thank the reviewer for pointing this out. The citations and phrasing in sections 4.3.1 and 4.3.2 have been updated to specifically refer to the weakly electric fish e. Virescens.

(3) Table on page 21: Please explain why the parameter value (13.5mM) of [Na^^^+]_{in} is 10 timeslarger than its value (1.35mM) in reference [26]? How does this value (13.5mM) compare with the range of variable [Na^^^+]_{in} in equation (6)?

The intracellular sodium concentration in reference [26] was reported to be 1.35 mM, but the authors also reported an extracellular sodium concentration of 120 mM, and a sodium reversal potential of 55 mV. Upon calculating the sodium reversal potential, we found that an intracellular sodium concentration of 1.35 mM would give a sodium reversal potential of 113 mV. An intracellular sodium concentration of 13.5 mM, on the other hand, leads to the reported and physiological reversal potential of 55 mV. This has now been clarified in the article, and the connection between this value and Eq. 6 (Eq. 7 in the new version) has also been clarified.

**Reviewer #2 (Public review):**
Summary:The paper by Weerdmeester, Schleimer, and Schreiber uses computational models to present the biological constraints under which electrocytes - specialized, highly active cells that facilitate electro-sensing in weakly electric fish-may operate. The authors suggest potential solutions that these cells could employ to circumvent these constraints.Electrocytes are highly active or spiking (greater than 300Hz) for sustained periods (for minutes to hours), and such activity is possible due to an influx of sodium and efflux of potassium ions into these cells after each spike. The resulting ion imbalance must be restored, which in electrocytes, as with many other biological cells, is facilitated by the Na-K pumps at the expense of biological energy, i.e., ATP molecules. For each ATP molecule the pump uses, three positively charged sodium ions from the intracellular space are exchanged for two positively charged potassium ions from the extracellular space. This creates a net efflux of positive ions into the extracellular space, resulting in hyperpolarized potentials for the cell over time. For most cells, this does not pose an issue, as their firing rate is much slower, and other compensatory mechanisms and pumps can effectively restore the ion imbalances. However, in the electrocytes of weakly electric fish, which spike at exceptionally high rates, the net efflux of positive ions presents a challenge. Additionally, these cells are involved in critical communication and survival behaviors, underscoring their essential role in reliable functioning.In a computational model, the authors test four increasingly complex solutions to the problem of counteracting the hyperpolarized states that occur due to continuous NaK pump action to sustain baseline activity. First, they propose a solution for a well-matched Na leak channel that operates in conjunction with the NaK pump, counteracting the hyperpolarizing states naturally. Their model shows that when such an orchestrated Na leak current is not included, quick changes in the firing rates could have unexpected side effects. Secondly, they study the implications of this cell in the context of chirps-a means of communication between individual fish. Here, an upstream pacemaking neuron entrains the electrocyte to spike, which ceases to produce a so-called chirp - a brief pause in the sustained activity of the electrocytes. In their model, the authors demonstrate that including the extracellular potassium buffer is necessary to obtain a reliable chirp signal. Thirdly, they tested another means of communication in which there was a sudden increase in the firing rate of the electrocyte, followed by a decay to the baseline. For this to occur reliably, the authors emphasize that a strong synaptic connection between the pacemaker neuron and the electrocyte is necessary. Finally, since these cells are energy-intensive, they hypothesize that electrocytes may have energy-efficient action potentials, for which their NaK pumps may be sensitive to the membrane voltages and perform course correction rapidly.Strengths:The authors extend an existing electrocyte model (Joos et al., 2018) based on the classical Hodgkin and Huxley conductance-based models of sodium and potassium currents to include the dynamics of the sodium-potassium (NaK) pump. The authors estimate the pump's properties based on reasonable assumptions related to the leak potential. Their proposed solutions are valid and may be employed by weakly electric fish. The authors explore theoretical solutions to electrosensing behavior that compound and suggest that all these solutions must be simultaneously active for the survival and behavior of the fish. This work provides a good starting point for conducting in vivo experiments to determine which of these proposed solutions the fish employ and their relative importance. The authors include testable hypotheses for their computational models.Weaknesses:The model for action potential generation simplifies ion dynamics by considering only sodium and potassium currents, excluding other ions like calcium. The ion channels considered are assumed to be static, without any dynamic regulation such as post-translational modifications. For instance, a sodium-dependent potassium pump could modulate potassium leak and spike amplitude (Markham et al., 2013).This work considers only the sodium-potassium (NaK) pumps to restore ion gradients. However, in many cells, several other ion pumps, exchangers, and symporters are simultaneously present and actively participate in restoring ion gradients. When sodium currents dominate action potentials, and thus when NaK pumps play a critical role, such as the case in Eigenmannia virescens, the present study is valid. However, since other biological processes may find different solutions to address the pump's non-electroneutral nature, the generalizability of the results in this work to other fast-spiking cell types is limited. For example, each spike could include a small calcium ion influx that could be buffered or extracted via a sodium-calcium exchanger.

We thank the reviewer for the detailed summary and the updated identified strengths and weaknesses. The current article indeed focuses on and isolates the interplay between sodium currents, potassium currents, and sodium-potassium pump currents. As discussed in section 5.1, in excitable cells where these currents are the main players in action-potential generation, the results presented in this article are applicable. The contribution of post-translational effects of ion channels, other ionic currents, and other active transporters and pumps, could be exciting avenues for further studies

.

**Reviewer #2 (Recommendations for the authors):**
Thank you for addressing my comments.All the figures are now consistent. The color schema used is clear.The methods and discussions expansions improve the paper.Including the model assumptions and simplifications is appreciated.Including internal references is helpful.The equations are clear, and the references have been fixed.I am content with the changes. I have updated my review accordingly.

We thank the reviewer for their initial constructive comments that lead to the significant improvement of the article.

Page : 3 Line : 113 Author : Unknown Author 07/24/2025Although this is technically correct, the article is about electrocommunication signals and does not focus on sensing.Page : 3 Line : 153 Author : Unknown Author 07/24/2025electrocommunicationPage : 4 Line : 164 Author : Unknown Author 07/24/2025Judging from the cited article, I think this should be a sodium-dependent potassium current.